# Atmospheric oxidation capacity and ozone pollution mechanism in a coastal city of Southeast China: Analysis of a typical photochemical episode by Observation-Based Model

Taotao Liu[1,2,3], Youwei Hong[1,2], Mengren Li[1,2], Lingling Xu[1,2], Jinsheng Chen[1,2]*, Yahui Bian[1,2], Chen Yang[1,2,3], Yangbin Dan[1,2], Yingnan Zhang[4], Likun Xue[4]*, Min Zhao[4], Zhi Huang[5], Hong Wang[6]

[1]Center for Excellence in Regional Atmospheric Environment, Institute of Urban Environment, Chinese Academy of Sciences, Xiamen, China

[2]Key Lab of Urban Environment and Health, Institute of Urban Environment, Chinese Academy of Sciences, Xiamen, China

[3]University of Chinese Academy of Sciences, Beijing, China

[4]Environment Research Institute, Shandong University, Jinan, Shandong, China

[5]Xiamen Institute of Environmental Science, Xiamen, China

[6]Fujian Meteorological Science Institute, Fujian Key Laboratory of Severe Weather, Fuzhou, China

Corresponding authors E-mail: Jinsheng Chen (jschen@iue.ac.cn); Likun Xue (xuelikun@sdu.edu.cn)

## Abstract:

A typical multi-day ozone ($O_3$) pollution event was chosen to explore the atmospheric oxidation capacity (AOC), OH reactivity, radical chemistry, and $O_3$ pollution mechanism in a coastal city of Southeast China, with an Observation-Based Model coupled to the Master Chemical Mechanism (OBM-MCM). The hydroxyl radical (OH) was the predominant oxidant ($90\pm25\%$) for daytime AOC, while $NO_3$ radical played an important role for AOC during the nighttime ($72\pm9\%$). Oxygenated volatile organic compounds (OVOCs, $30\pm8\%$), $NO_2$ ($29\pm8\%$) and CO ($25\pm5\%$) were the dominant contributors to OH reactivity, accelerating the production of $O_3$ and recycling of ROx radicals (ROx=OH+HO$_2$+RO$_2$). Photolysis of nitrous acid (HONO, $33\pm14\%$), $O_3$ ($25\pm13\%$), formaldehyde (HCHO, $20\pm5\%$), and other OVOCs ($17\pm2\%$) were major ROx sources, which played initiation roles in atmospheric oxidation processes. Combined with regional transport analysis, the reasons for this $O_3$ episode were the accumulation of local photochemical production and regional transport. The results of sensitivity analysis showed that VOCs were the limiting factor of radical recycling and $O_3$ formation, and the 5% reduction of $O_3$ would be achieved by decreasing 20% anthropogenic VOCs, and controlling emissions of aromatics, alkenes, and alkanes with $\geq4$ carbons were benefit for ozone pollution mitigation. The findings of this study have significant guidance for emission reduction and regional collaboration on future photochemical pollution control in the relatively clean coastal cities of China and similar countries.

**Keywords**: Atmospheric oxidation capacity; Radical chemistry; $O_3$ formation mechanism; OH reactivity;

OBM-MCM

**1 Introduction**

Tropospheric ozone ($O_3$) is mainly produced by photochemical reactions of anthropogenic and
natural emitted volatile organic compounds (VOCs) and nitrogen oxides (NOx), and is an important factor
resulting in regional air pollution (Zhu et al., 2020; Lu et al., 2018). The elevated $O_3$ concentrations
enhance the atmospheric oxidation capacity (AOC) and have harmful effects on global climate change,
ecosystems, and human health (Liu et al., 2019a; Fowler et al., 2009). The formation mechanisms of $O_3$
pollution are extremely difficult to figure out, due to the complex types and sources of its precursors
(Simon et al., 2015). $O_3$ formation is affected by multiple factors such as $O_3$ precursor speciation or level,
atmospheric oxidation capacity, meteorological conditions and regional transport (Gong and Liao, 2019;
Chang et al., 2019). To effectively control the tropospheric $O_3$ pollution, exploration of the photochemical
mechanism and judgment on the controlling factors of $O_3$ formation become extremely important for
scientific community (Chen et al., 2020; Li et al., 2018).
The atmospheric oxidation capacity reflects the essential driving force in tropospheric chemistry,
and plays an important place in the loss rates of primary components and production rates of secondary
pollutants, thus the key factors to quantify AOC are processes and rates of species being oxidized in the
atmosphere (Elshorbany et al., 2009). The atmospheric conditions (such as photolysis rate, meteorology,
pollutant concentrations and regional transport) together influence the AOC levels, and the AOC levels
in the polluted urban regions are generally much higher than those at the background sites or remote
regions due to the dominant limited factor for the significant differences of pollutant concentrations(Geyer
et al., 2001; Xue et al., 2016). ROx radicals, including hydroxyl radical (OH), hydro peroxy radical ($HO_2$)
and organic peroxy radical ($RO_2$), are very important indicators in atmospheric photochemistry and
dominate the atmospheric oxidation capacity (Li et al., 2018). Meanwhile, radical chemistry drives the
transformation and recycling of $O_3$ through initiating atmospheric oxidation processes (Wang et al., 2020).
Among these radicals, the OH radical accounts for the majority of AOC over 90% during the daytime,
thus the OH reactivity (i.e., OH loss) indicates the primary contribution of individual pollutants (Wang et
al., 2018a; Mao et al., 2010). Hence, atmospheric oxidation capacity, OH reactivity, and radical chemistry
are crucial aspects for understanding the complex atmospheric photochemistry processes (Li et al., 2018).
For example, the major ROx sources are the photolysis reaction of $O_3$, formaldehyde (HCHO), other
oxygenated volatile organic compounds (OVOCs), nitrous acid (HONO) and the reactions of $O_3$ with
unsaturated VOCs (Volkamer et al., 2010). The dominant ROx sources at some rural sites were $O_3$
photolysis and $O_3$ reactions with VOCs (Li et al., 2018; Martinez et al., 2003), and those at many urban
sites were HONO and OVOCs photolysis (Xue et al., 2016; Liu et al., 2012; Emmerson et al., 2005). For
oil and gas field sites, there were highly abundant VOCs to promote the formations of $O_3$, and the
contribution of OVOCs photolysis was 2-5 times higher than that in urban areas (Chen et al., 2020;
Edwards et al., 2013, 2014). The HONO photolysis was a very important ROx source at the high-altitude
or background sites. (Acker et al., 2001; Jiang et al., 2020).
Current studies of atmospheric $O_3$ photochemical pollution observations have been conducted at the
urban, suburban, rural and remote sites around the world (Smith et al., 2006; Eisele et al., 1997; Kanaya
et al., 2001; Hofzumahaus et al., 2009; George et al., 1999; Emmerson et al., 2005; Kanaya et al., 2007;
Michoud et al., 2012). In China, $O_3$ photochemical pollution events have been reported in some megacities,
such as Beijing, Shanghai, Guangzhou, and Chengdu (Liu et al., 2012; Tan et al., 2019; Zhu et al., 2020;
Wang et al., 2020; Liu et al., 2019b; Ling et al., 2017). Few studies on $O_3$ photochemical pollution in
cities with low $O_3$ precursor emissions have been reported, and the air quality in these areas usually
depends on the change of meteorological conditions. In a coastal city of Southeast China, the
concentrations of $O_3$ precursors were higher than those in remote sites and background, but lower than
those in most urban and suburban areas, even lower than those in rural regions (Table S1). In a word, $O_3$
precursor emissions in our observation site were relatively low. Meanwhile, the southeast coastal region
is influenced by the East Asian monsoon and acts as an important transport path between the Yangtze
River Delta (YRD) and the Pearl River Delta (PRD) (Liu et al., 2020a; Liu et al., 2020b), which is a good
'laboratory' to further explore $O_3$ photochemical pollution and formation mechanism with relatively low
$O_3$ precursors and complex meteorological conditions (Zhang et al., 2020b; Hu et al., 2020).
The Observation-Based Model (OBM) is widely used to investigate $O_3$-VOCs-NOx relationships
and radical chemistry (Wang et al., 2018a; Tan et al., 2019). The $O_3$ sensitivity revealed the non-linear
relationship between $O_3$ and its precursors (i.e., VOCs and NOx), which was conducted to investigate $O_3$
formation mechanism and control strategies (Wang et al., 2020). The OBM combined with the Master
Chemical Mechanism (V3.3.1) (OBM-MCM) has been applied to explore the $O_3$ photochemical pollution
mechanism in different environmental conditions (Chen et al., 2020; Li et al., 2018; Xue et al., 2016;
Wang et al., 2018). In this study, we chose a typical multi-day $O_3$ pollution event in the coastal city Xiamen
(Fig. S1), when Xiamen was affected by various meteorological conditions, such as typhoon and the West
Pacific Subtropical High (WPSH) accompanied by temperature inversion phenomenon. Based on the
OBM-MCM analyses, the study aims to clarify (1) the pollution characteristics of $O_3$ and its precursors,
(2) the atmospheric oxidation capacity and radical chemistry, and (3) the $O_3$ formation mechanism and
sensitivity analysis. The results are expected to enhance the understanding of the $O_3$ formation mechanism
with low $O_3$ precursor levels, and provide scientific evidence for $O_3$ pollution control in the coastal cities.

**2 Materials and methods**
**2.1 Study area and field observations**
Xiamen is a coastal city in the southeast area of China, to the west coast of the Taiwan Strait. The
field campaigns were carried out at the Atmospheric Environment Observation Supersite (24.61° N,
118.06° E) on the rooftop of around 70 m high building in the Institute of Urban Environment, Chinese
Academy of Sciences. The supersite was equipped with complete monitoring instruments, including gas
and aerosol species compositions, $O_3$ precursors, meteorological parameters, and photolysis rate. Criteria
air pollutants of $O_3$, $SO_2$, NO-$NO_2$-NOx, and CO were monitored by commercial instruments TEI 49i,
43i, 42i, and 48i (*Thermo Fisher Scientific, USA*), respectively. The meteorological parameters of wind
speed (WS), wind direction (WD), air temperature (T), pressure (P), and relative humidity (RH) were
measured by an ultrasonic atmospherium (*150WX, Airmar, USA*). HONO was measured with an analyzer
for Monitoring Aerosols and Gases in Ambient Air (*MARGA, ADI 2080, Applikon Analytical B.V., the*
*Netherlands*). A gas chromatography-mass spectrometer (GC-FID/MS, *TH-300B, Wuhan, CN*) was used
for atmospheric VOCs concentrations monitoring, involving about 103 species of VOCs with a 1-hour
time resolution. Photolysis frequencies were measured by a photolysis spectrometer (*PFS-100, Focused*
*Photonics Inc., Hangzhou, China*). The photolysis rate constants include $J(O^1D)$, $J(NO_2)$, $J(H_2O_2)$,
$J(HONO)$, $J(HCHO)$, and $J(NO_3)$. Strict quality assurance and quality control were applied, and the
detailed descriptions of the monitoring procedures were documented in our previous studies (Zhang et al.,
2020b; Wu et al., 2020; Liu et al., 2020a; Liu et al., 2020b; Hu et al., 2020).
**2.2 Observation-based chemical box model**
In this study, the Observation-Based Model (OBM) combined with the latest version 3.3.1 of MCM
(MCM v3.3.1; http://mcm.leeds.ac.uk/MCM/), involving 142 non-methane VOCs and more than 17000
elementary reactions of 6700 primary, secondary and radical species (Jenkin et al., 2003; Saunders et al.,
2003), was used to explore the atmospheric oxidation processes and $O_3$ formation mechanisms. The
physical process of deposition within the boundary layer height (BLH), which varied from 300 m during
nighttime to 1500 m during the daytime in autumn (Li et al., 2018), was considered in the model.
Therefore, the dry deposition velocity was utilized to simulate the deposition loss of some reactants in the
atmosphere and showed in Table S2, which avoided continuous accumulation of pollutant concentrations
in the model (Zhang et al., 2003; Xue et al., 2016).

133  The observation parameters of the gaseous pollutants (i.e., $O_3$, CO, NO, $NO_2$, HONO, $SO_2$, and

134  VOCs), meteorological parameters (i.e., T, P, and RH), and photolysis rate constants ($J(O^1D)$, $J(NO_2)$,

135  $J(H_2O_2)$, $J(HONO)$, $J(HCHO)$, and $J(NO_3)$)) were input into the OBM-MCM model as constraints. The

136  photolysis rates of other molecules such as OVOCs were parameterized by solar zenith angle and then

137  scaled by the measured $J(NO_2)$ (Saunders et al., 2003). We pre-ran for 5 days before running the model

138  to initialize the unmeasured compounds and radicals (Xue et al., 2014).

139  OBM-MCM is mainly used to simulate in situ atmospheric photochemical processes and quantify

140  the $O_3$ production rate, AOC, OH reactivity, and ROx radical budgets. Among them, primary sources of

141  ROx, including the photolysis reactions of $O_3$, HONO, formaldehyde (HCHO), and other OVOCs as well

142  as reactions of VOCs with $O_3$ and $NO_3$ radicals, are important (Xue et al., 2016). The termination reactions

143  of ROx are controlled by cross-reactions with NOx (under high NOx conditions) and ROx (under low

144  NOx conditions) to form nitric acid, organic nitrates, and peroxides (Liu et al., 2012; Xue et al., 2016).

145  Table 1 shows the production and destruction reactions and relevant reaction rates of $O_3$ in our study. The

146  production rate of $O_3$ ($P(O_3)$) includes $RO_2$+NO (R1) and $HO_2$+NO reactions (R2, Eq. 1), and the

147  destruction of $O_3$ ($D(O_3)$) involves reactions of $O_3$ photolysis (R3), $O_3$+OH (R4), $O_3$+$HO_2$ (R5), $NO_2$+OH

148  (R6), $O_3$+VOCs (R7), and $NO_3$+VOCs (R8, Eq. 2). The net $O_3$ production rate ($Pnet(O_3)$) is calculated

149  by $P(O_3)$ minus $D(O_3)$ as equation 3.


151  **Table 1. Simulated production and destruction reactions and relevant reaction rates of $O_3$ in our study.**

| Reactions | Reaction rates | Number |
|---|---|---|
| **$O_3$ production pathways-$P(O_3)$** | | |
| $RO_2$+NO→RO+$NO_2$ | $2.7\times10^{-12}\times EXP(360/T)$ | R1 |
| $HO_2$ +NO→OH+$NO_2$ | $3.45\times10^{-12}\times EXP(270/T)$ | R2 |
| **$O_3$ loss pathways-$D(O_3)$** | | |
| $O_3$+hv→$O^1D$+$O_2$ | $JO^1D$ | R3a |
| $O^1D$+$H_2O$→OH | $2.14\times10^{-10}$ | R3b |
| $O_3$+OH→$HO_2$ | $1.70\times10^{-12}\times EXP(-940/T)$ | R4 |
| $O_3$+$HO_2$→OH | $2.03\times10^{-16}\times(T/300)^{4.57}\times EXP(693/T)$ | R5 |
| $NO_2$+OH→$HNO_3$ | $3.2\times10^{-30}\times9.7\times10^{18}\times P/T\times(T/300)^{-4.5}\times3.0^{-11}\times10^{\log10(0.41)}/(1+(\log(3.2^{-30}\times9.7E\times10^{18}\times P/T\times(T/300)^{-4.5}\times3.0^{-11}/(0.75-1.27\times(\log_{10}(0.14))^2)/(3.2^{-30}\times9.7E\times10^{18}\times P/T\times(T/300)^{-4.5}+3.0^{-11})$ | R6 |
| $O_3$+VOCs→Carbonyls+Criegee biradical | Kcons.1 | R7 |
| $NO_3$+VOCs→$RO_2$ | Kcons.2 | R8 |

152 Note: The reaction rates of Kcons.1 and Kcons.2 were constant. There were around 700 reactions of VOCs+$NO_3$/$O_3$,

153 and the relevant reaction rates were different, which can be obtained from this website http://mcm.leeds.ac.uk/MCM/.


155 $P(O_3) = k_1[HO_2][NO] + \sum(k_{2i}[RO_2][NO])$             (1)

$D(O_3) = k_3[O_1D][H_2O] + k_4[O_3][OH] + k_5[O_3][HO_2] + k_6[NO_2][OH] +$

157          $\sum(k_{7i}[O_3][unsat.VOCs]) + 2\sum(k_{8i}[NO_3][unsat.VOCs])$         (2)

$Pnet(O_3) = P(O_3) - D(O_3)$                         (3)
where $ki$ is the related reaction rate constant. Detailed descriptions of the chemistry calculation can be
found elsewhere (Chen et al., 2020; Wang et al., 2018a; Xue et al., 2014).

161       Relative incremental reactivity (RIR), an index to diagnose the sensitivity of $O_3$ formation to

precursors, is defined as the ratio of the differences in $O_3$ production rate to the difference in precursor
concentrations (Chen et al., 2020). Here, the $\Delta X/X$ in the OBM-MCM represents the percentage reduction
in the input concentrations of each targeted $O_3$ precursor group and this value is adopted as 20% (Liu et
al., 2020c).
$RIR = \frac{\Delta P(O_3)/P(O_3)}{\Delta X/X}$                               (4)

**2.3 Model performance**

169       The index of agreement (IOA) can be used to judge the reliability of the model simulation results,

and its equation is (Liu et al., 2019b):
$IOA = 1 - \frac{\sum_{i=1}^{n}(O_i - S_i)^2}{\sum_{i=1}^{n}(|O_i - \bar{O}| - |S_i - \bar{O}|)^2}$             (5)

172       where $Si$ is simulated value, $Oi$ represent observed value, $\bar{O}$ the average observed values, and n is

the sample number. The IOA range is 0-1, and the higher the IOA value is, the better agreement between
simulated and observed values is. In many studies, when IOA ranges from 0.68 to 0.89 (Wang et al.,
2018b), the simulation results are reasonable, and the IOA in our research is 0.80. Hence, the performance
of the OBM-MCM model was reasonably acceptable.

**2.4 Meteorological data and back trajectory calculation**

179       The backward trajectories of air masses arriving at the observation site were calculated by the

MeteoInfo during the episode (Wang *et al.*, 2014). The backward trajectories with 72-h were run with the
time resolution of 3 hours at 100 m height above ground level, and starting time was 0:00 LT and the
ending time was 23:00 LT. Meteorological data were provided by NOAA ARL
(ftp://arlftp.arlhq.noaa.gov/pub/archives/gdas1). The Final Operational Global Analysis data (FNL) is
from the Global Data Assimilation System and analyzes results with the model which is also used by the
National Center for Environmental Prediction (NCEP) in the Global Forecast System (GFS)

. The weather charts were conducted using Grid Analysis and Display System (GrADS) with the specific programmed script files. A detailed description of the synoptic information was shown in our previous study (Wu et al., 2019).

## 3 Results and discussion

### 3.1 Overview of observations

The $O_3$ pollution events frequently appeared in the coastal city Xiamen during autumn time, related to the WPSH, carrying favorable photochemical reaction conditions (high temperature, low RH, and stagnant weather conditions) and encouraging the formation and accumulation of $O_3$ in the southeast coastal area (Wang et al., 2018a). The daily maximum 8-h-average $O_3$ concentrations (MDA8h $O_3$) from 20 to 29 Sep 2019 ranged from 53 to 85 ppbv, partly exceeding the Grade II of China's National Ambient Air Quality Standard of 75 ppbv. The time series and descriptive statistics of air pollutants and meteorological parameters during this multi-day $O_3$ pollution event are shown in Fig. 1 and Table 2. During this period, the dominant wind direction was northeast, with an average wind speed of $1.8\pm0.9$ $m\cdot s^{-1}$. The maximum hourly temperature was as high as 35 °C, and the average RH was $56.4\pm12.6\%$. Our previous study showed that particulate pollution was slight in Xiamen, which could affect solar radiation by light-absorbing component, and the concentrations of particulate matter had not exceeded the National Ambient Air Quality Standard (Class II: 75 $\mu g\cdot m^{-3}$) for a whole year (Hu et al., 2021; Deng et al., 2020). Therefore, solar radiation intensity and $J(NO_2)$ were strong, compared to those of the Yellow River Delta (Chen et al., 2020), Shanghai (Zhu et al., 2020) and Hong Kong (Xue et al., 2016). In general, these meteorological parameters were conducive to the production and accumulation of $O_3$. In addition, $O_3$ concentrations at nighttime kept relatively high (Fig.1), indicating the influence of regional transport and little NO titration (Zhang et al., 2020a; Wu et al., 2020). Figure S2 shows the 72 h back trajectories at the monitoring site. Among them, 80% of the air masses came from the Yellow Sea, and the other 20% air masses originated from the northeast China through long-range transport.

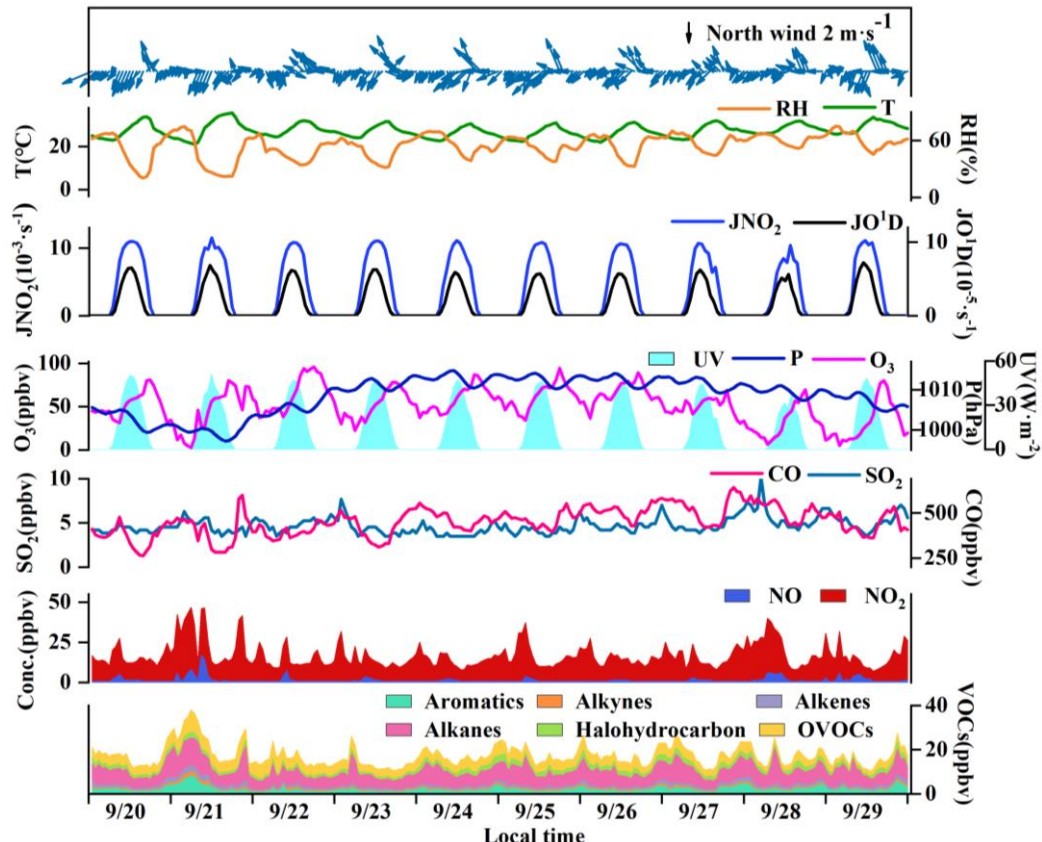

210

Figure 1. Time series of major trace gases, photolysis rate constants, and meteorological parameters during 20-29 Sep. 2019 in Xiamen.

213

Table 2. Descriptive statistics of major trace gases (ppbv) and meteorological parameters during 20-29 Sep. 2019.

| Parameters | Mean±SD | Median | Max |
|---|---|---|---|
| MDA8h $O_3$ | 67.4±17.2 | 52.6 | 89.3 |
| TVOCs | 17.2±4.8 | 16.1 | 38.0 |
| CO | 452±77.3 | 454 | 641 |
| NO | 1.4±1.3 | 0.8 | 17.1 |
| $NO_2$ | 15.4±6.9 | 13.6 | 40.9 |
| $SO_2$ | 4.7±0.9 | 4.6 | 10.2 |
| T (℃) | 27.3±3.21 | 26.9 | 35.6 |
| RH (%) | 56.4±12.6 | 56.6 | 75.0 |
| P (hPa) | 1008±4.57 | 1010 | 1015 |
| UV (W·m$^{-2}$) | 46.4±1.12 | 0 | 51.1 |
| Wind speed (m·s$^{-1}$) | 1.8±0.9 | 1.6 | 3.8 |
| Wind direction (∘) | 90.8±90.4 | 45.0 | 337 |

Table 3. Measured VOCs concentrations during 20-29 Sep. 2019 in Xiamen (Units: pptv), and the classification of VOCs were used and introduced in Section 3.3.

| Chemicals | Classification | Mean±SD | Chemicals | Classification | Mean±SD |
|---|---|---|---|---|---|
| **Aromatics** | | **2131±1236** | **Alkanes** | | **6970±2325** |
| toluene | RAROM/AHC | 995±632 | ethane | LRHC/AHC | 1552±342 |
| m/p-xylene | RAROM/AHC | 392±326 | propane | LRHC/AHC | 1546±608 |
| benzene | LRHC/AHC | 236±95 | iso-pentane | C4HC/AHC | 930±316 |
| o-xylene | RAROM/AHC | 154±121 | n-butane | C4HC/AHC | 844±365 |
| ethylbenzene | RAROM/AHC | 138±94 | n-dodecane | C4HC/AHC | 618±101 |
| styrene | RAROM/AHC | 76±65 | iso-butane | C4HC/AHC | 494±201 |
| 1,2,4-trimethylbenzene | RAROM/AHC | 75±37 | n-pentane | C4HC/AHC | 254±157 |

| | | | | | |
|---|---|---|---|---|---|
| m-ethyltoluene | RAROM/AHC | 16±11 | n-hexane | C4HC/AHC | 134±184 |
| p-ethyltoluene | RAROM/AHC | 10±6 | 3-methylhexane | C4HC/AHC | 116±93 |
| iso-propylbenzene | RAROM/AHC | 5±3 | n-heptane | C4HC/AHC | 104±78 |
| 1,3,5-trimethylbenzene | RAROM/AHC | 8±6 | 3-methylpentane | C4HC/AHC | 82±48 |
| o-ethyltoluene | RAROM/AHC | 8±5 | 2-methylhexane | C4HC/AHC | 67±38 |
| 1,2,3-trimethylbenzene | RAROM/AHC | 7±5 | 2-methylpentane | C4HC/AHC | 56±46 |
| n-propylbenzene | RAROM/AHC | 7±4 | 2,3-dimethylbutane | C4HC/AHC | 54±33 |
| **Halocarbons** | | **1951±572** | cyclohexane | C4HC/AHC | 42±15 |
| dichloromethane | AHC | 998±392 | n-undecane | C4HC/AHC | 33±35 |
| 1,2-dichloroethane | AHC | 499±210 | n-octane | C4HC/AHC | 24±15 |
| chloromethane | AHC | 294±75 | n-nonane | C4HC/AHC | 15±13 |
| 1,2-dichloropropane | AHC | 88±34 | 2,2-dimethylbutane | C4HC/AHC | 15±7 |
| bromomethane | AHC | 47±23 | n-decane | C4HC/AHC | 14±11 |
| trichloroethene | AHC | 15±6 | **Alkenes** | | **1205±464** |
| 1,4-dichlorobenzene | AHC | 9±3 | ethene | Alkenes/AHC | 671±361 |
| **OVOCs** | AHC | **4246±1263** | propene | Alkenes/AHC | 207±116 |
| acetone | AHC | 2802±750 | isoprene | BHC | 171±232 |
| 2-butanone | AHC | 799±430 | trans-2-pentene | Alkenes/AHC | 105±62 |
| 2-propanol | AHC | 343±283 | 1-butene | Alkenes/AHC | 16±21 |
| 2-methoxy-2-methylpropane | AHC | 169±97 | cis-2-butene | Alkenes/AHC | 12±12 |
| acrolein | AHC | 66±22 | 1-pentene | Alkenes/AHC | 10±7 |
| 4-methyl-2-pentanone | AHC | 16±15 | 1,3-butadiene | Alkenes/AHC | 8±7 |
| 2-hexanone | AHC | 12±3 | trans-2-butene | Alkenes/AHC | 4±4 |
| | | | **Acetylene** | LRHC/AHC | **674±290** |

Table 3 lists the detailed VOCs concentrations during the observation period. Alkanes (6970±2325 pptv) were the predominant components of total VOCs, followed by OVOCs (4246±1263 pptv), aromatics (2131±1236 pptv), halocarbons (1951±572 pptv), alkenes (1205±464 pptv), and acetylene (674±290 pptv). The ratio of ethene/ethane (0.4±0.2) was significantly (p<0.05) lower than that in Hong Kong (0.7±0.1) with significant aged air masses, indicating that the high $O_3$ in Xiamen might be partially attributed to the aged air masses (e.g., transport of air from polluted regions or intense atmospheric oxidation) (Wang et al., 2018a). The concentration of TVOCs in Xiamen (17.2±4.8 ppbv) was much lower than that in the developed areas with large anthropogenic emissions (i.e., Beijing (44.2 ppbv), Lanzhou (45.3 ppbv), Wuhan (30.2 ppbv), Chengdu (36.0 ppbv), Hong Kong (26.9 ppbv), Los Angeles (41.3 ppbv) and Tokyo (43.4 ppbv), but was higher than that at the background and remote sites (i.e., Mt. Wuyi 4.7 ppbv and Mt. Waliguan 2.6 ppbv) (Table S1).

The $O_3$ formation process depends on its precursors and related environmental conditions, while the photochemical reactions during the daytime are the basis for $O_3$ changes. Figure 2 shows the diurnal patterns of major trace gases and meteorological parameters during 20-29 Sep. 2019. The $O_3$ concentration was maintained at relatively low levels from night to 07:00 LT, then rose and reached its maximum at around 17:00 LT. $O_3$ peak in the afternoon was related to the accumulation of both local

photochemical reaction and potential regional transport (including $O_3$ and its precursors in the upwind
direction to the observation site), and the detailed analysis will be shown in Section 3.3.2. The reduction
of observed $O_3$ ($\Delta O_3$) in the early morning rush hour caused by NO titration did not appear, verifying the
impacts of regional transport (Liu et al., 2019b; Zeren et al., 2019; Chen et al., 2020). Due to the
photochemical reactions, the precursors of CO, NOx and VOCs were consumed during the daytime, and
were accumulated during the nighttime with weak solar radiation. The diurnal patterns of VOCs, NOx
and CO were similar, with the highest concentrations at around 08:00 LT and then decreasing during
9:00~16:00 LT and increasing at night, which is related to the human activity emissions (especially
vehicle exhaust) and the variations of boundary layer (Elshorbany et al., 2009; Hu et al., 2020).

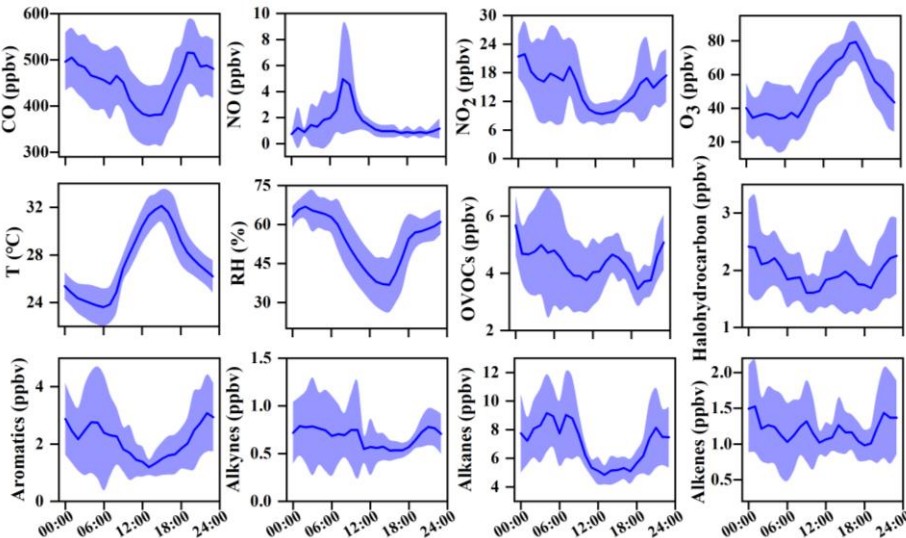


**Figure 2. Average diurnal patterns of major trace gases and meteorological parameters during 20-29 Sep. 2019 in Xiamen. The error bar is the standard error.**


**3.2 Atmospheric oxidation and radical chemistry**

**3.2.1 Atmospheric oxidation capacity (AOC)**



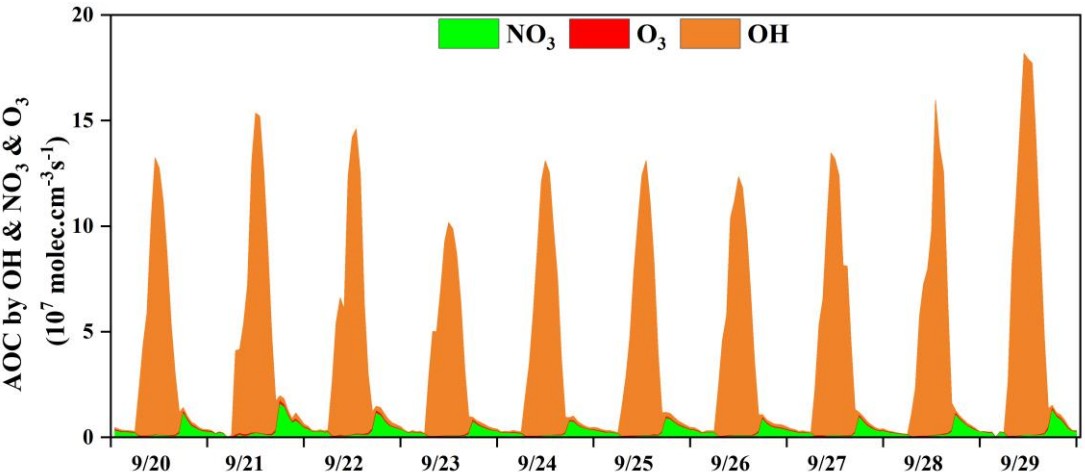

**Figure 3. Time series of the model-calculated Atmospheric Oxidation Capacity (AOC) in Xiamen during 20-29 Sep. 2019.**

Figure 3 shows the time series of the model-calculated AOC during the $O_3$ pollution period. The AOC determines the removal rate of primary pollutants and the production rate of secondary pollutants, and was the basis for reflecting atmospheric photochemical pollution (Geyer et al., 2001). AOC is calculated as the sum of oxidation rates of various primary pollutants (CO, NOx, VOCs, etc.) by the major oxidants (i.e., OH, $O_3$, $NO_3$) (Chen et al., 2020; Xue et al., 2016; Xue et al., 2014). In this study, the average daytime AOC was $6.7×10^7$ molecules $cm^{-3}$ $s^{-1}$, and the daily maximum AOC was $1.3×10^8$ molecules $cm^{-3}$ $s^{-1}$, which was higher than those at rural sites with much low pollution emissions in Berlin ($1.4×10^7$ molecules $cm^{-3}$ $s^{-1}$) and a regional background in Hong Kong ($6.2 × 10^7$), but lower than that in polluted cities, such as Santiago ($3.2×10^8$ molecules $cm^{-3}$ $s^{-1}$), due to the main limited factor of the significant differences of pollutant concentrations among different sites (Li et al., 2018; Xue et al., 2016; Geyer et al., 2001; Zhu et al., 2020). In some urban regions, the concentrations of air pollutants were higher than those in Xiamen, but their AOC levels (Hong Kong: $1.3×10^8$ molecules $cm^{-3}$ $s^{-1}$; Shanghai: $1.0×10^8$ molecules $cm^{-3}$ $s^{-1}$) were comparable to or even lower compared with the AOC in Xiamen, which could be attributed to the relatively high solar radiation (Xue et al., 2016; Zhu et al., 2020) (Detailed descriptions showed in Section 3.1). The results of AOC characteristics in different regions were decided by the precursor concentrations/types and photochemical environment.

According to the diurnal patterns of the AOC contributed by OH, $O_3$, and $NO_3$, the predominant oxidant was OH (90±25%) during the daytime, followed by $NO_3$ (8±22%) and $O_3$ (2±3%). Meanwhile, the diurnal characteristics of AOC were consistent with the profile of the model-calculated OH (Fig. S3) and the observed photolysis rate constants (Fig.1) (Zhu et al., 2020). Meanwhile, $NO_3$ (72±9%) played the most important role in the oxidant capability during the nighttime, followed by OH (20±12%) and $O_3$

(8±1%). In particular, the contribution of NO$_3$ to AOC reached the maximum of 80% at around 18:00 LT,
when the concentrations of O$_3$ and NO$_2$ were relatively high and accelerated the formation of NO$_3$ (Fig.2).
In addition, solar radiation was weak during the nighttime, which resulted in the accumulation of NO$_3$
due to the cease of photolysis of NO$_3$ (Rollins et al., 2012; Chen et al., 2020). AOC contributed by O$_3$
was negligible, owing to the relatively low concentration of alkenes at the monitoring site (Fig.1 and
Table 3), since O$_3$ contributed to the oxidation capacity through alkenes ozonolysis (Xue et al., 2016). In
summary, the OH radical dominated the AOC, and it was necessary to further explore the partitioning of
OH reactivity among different precursor groups.

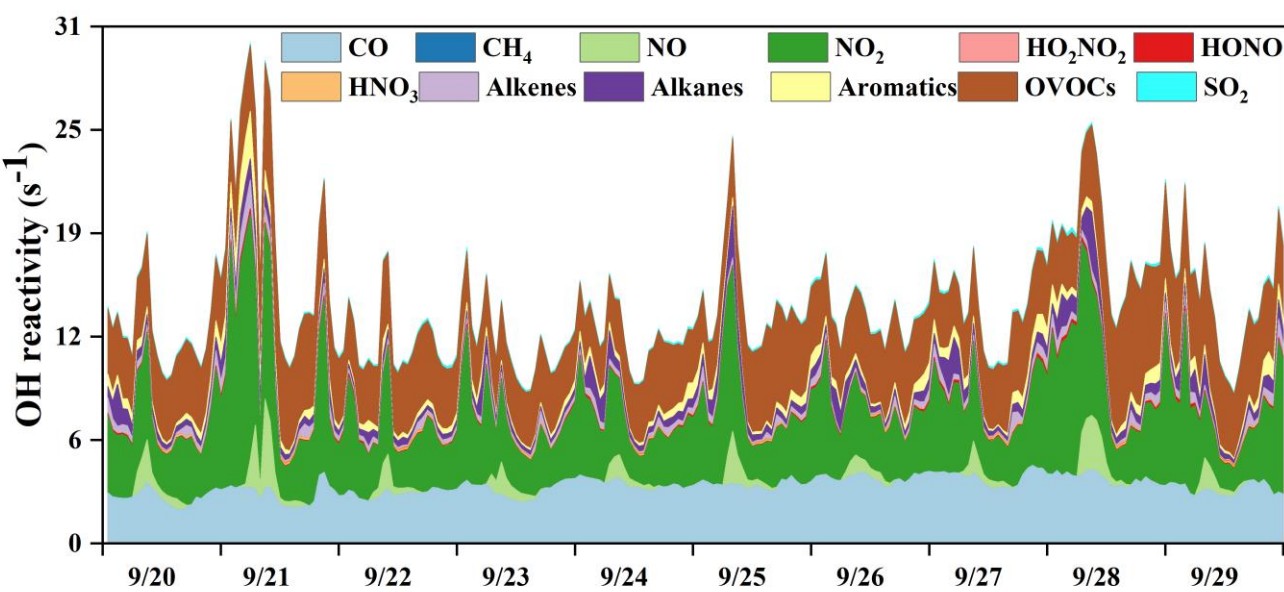


**Figure 4. Time series of model-calculated OH reactivity and its partitioning to the major reactants in Xiamen**
**during 20-29 Sep. 2019.**

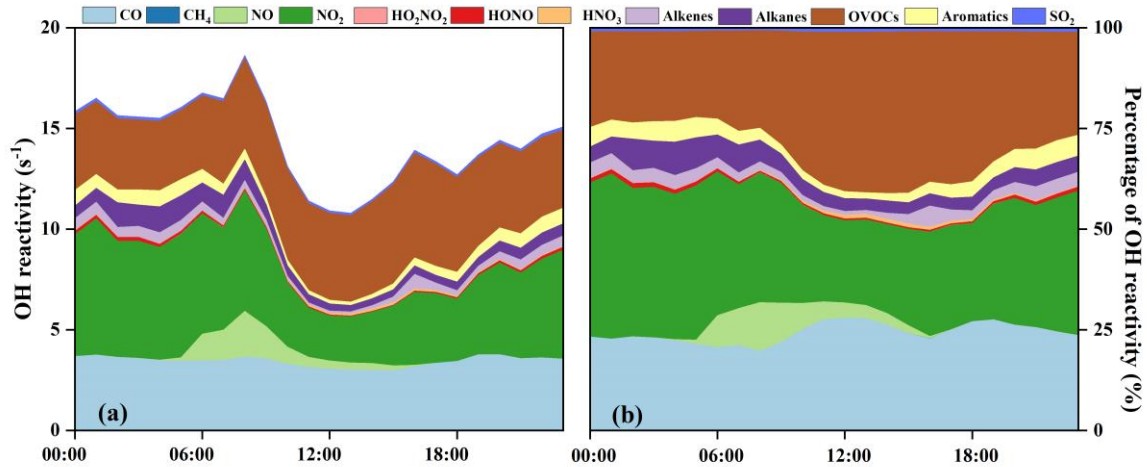

**Figure 5. (a) Diurnal patterns and (b) percentage of model-calculated OH reactivity and its partitioning to the**
**major reactants.**
The OH reactivity is an indicator for the OH chemical loss frequency, computed as the reaction rates
of OH with CO, NOx, $SO_2$, HONO, $HNO_3$, $HO_2NO_2$, and VOCs (Whalley et al., 2016; Chen et al., 2020).
Time series and diurnal patterns of model-calculated OH reactivity as well as its partitioning to the major
reactants during the episode are shown in Fig. 4 and Fig. 5. The OH reactivity reached the peak (18.6±4.8
$s^{-1}$) at around 8:00 LT, mainly caused by the reaction of OH with NOx, since vehicles exhausted large
amounts of NOx during rush hours. The average daily OH reactivity was 14.4±3.83 $s^{-1}$, which was much
lower than those in some polluted regions in Santiago (42 $s^{-1}$) and the PRD (50 $s^{-1}$), comparable to that
at a rural site in Nashville (11 $s^{-1}$), but higher than that at a mountain site in Pennsylvania (6 $s^{-1}$)
(Elshorbany et al., 2009; Lou et al., 2010a; Lou et al., 2010b; Kovacs et al., 2003; Ren et al., 2005). Figure
5 shows the diurnal variations and percentage of model-calculated OH reactivity to the major reactants
during the episode. The OH reactivity exhibited a morning peak caused by the reactions of NO with OH,
which should be ascribed to the freshly emitted urban plumes. Anymore, OVOCs showed high fractions
at around 12:00-18:00 LT, which were mainly owing to the transport of the regional air masses containing
the abundant OVOCs, as well as the oxidation effection by strong photochemical process. As shown in
Fig. 5b, OVOCs (30±8 %) , $NO_2$ (29±8%) and CO (25±5%), were the dominant contributors to OH
reactivity, followed by alkanes (5±3%), aromatics (3±2%), alkenes (3±1%), and NO (2±4%). The high
fraction of OVOCs and $NO_2$ in OH reactivity indicated the high aged degree of air mass and the intensive
$NO_x$ emissions during the observation period, respectively (Li et al., 2018). However, the fraction of CO
to OH reactivity at our observation site was higher than that at an urban site in Los Angeles (Hansen et
al., 2021), a rural site in Hong Kong (Li et al., 2018), and a mountain site in Colorado (Nakashima et al.,
2014), comparable to that at the urban site of Shanghai (Zhang et al., 2021a), which could be attributed
to the abundant CO in our observation site. CO mainly comes from vehicle exhaust and the combustion
of fossil fuels, and the observation site is a city with high density vehicles. Meanwhile, this pollution
event was under the influence of the WPSH, which promoted the formation and accumulation of
pollutants. The partitioning of OH reactivity elucidated the inherent photochemical processes and major
reactants in Southeast China. High OH reactivity of OVOCs, $NO_2$, and CO would promote the production
of ROx radical. Therefore, the investigation of detailed chemical budget of the ROx, recycling, and
termination reaction is meaningful to figure out the complex atmospheric photochemistry (Li et al., 2018;
Lou et al., 2010b).
**3.2.2 Radical chemistry**

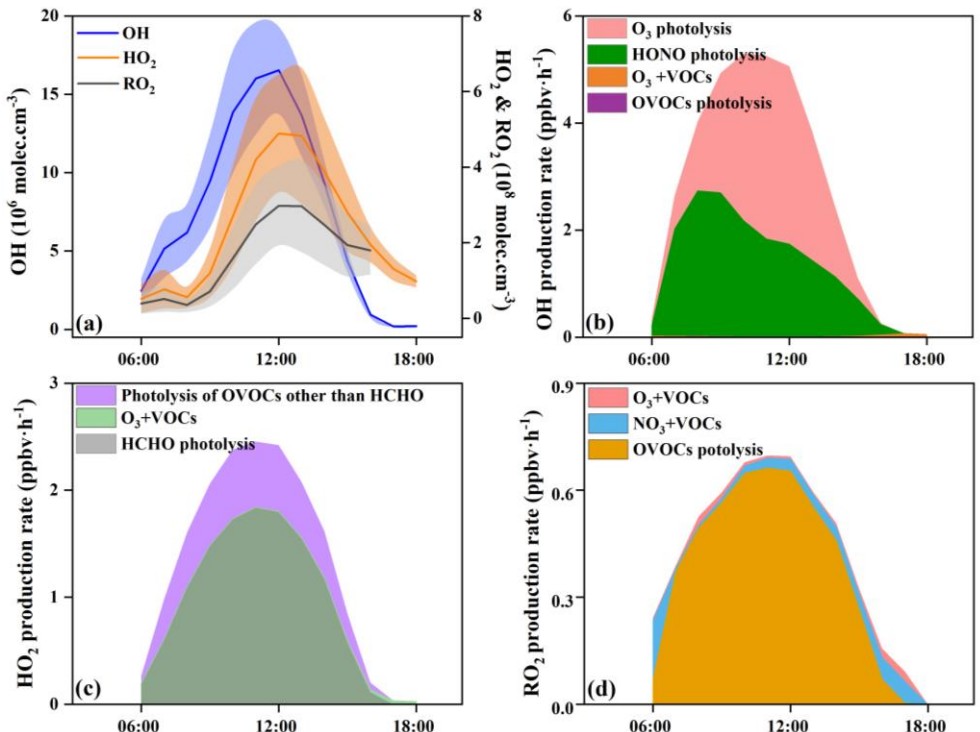

**Figure 6. Model-simulated daytime average diurnal variations in (a) OH, HO$_2$, and RO$_2$ concentrations, and average primary production rates of (b) OH, (c) HO$_2$, and (d) RO$_2$ during 20-29 Sep. 2019 in Xiamen.**

With the influence of NOx and VOCs, ROx radicals (OH, HO$_2$, and RO$_2$) undergo efficient recycling and produce secondary pollutants, such as O$_3$ and OVOCs (Sheehy et al., 2010). Figure 6 shows the model-simulated OH, HO$_2$, and RO$_2$ concentrations and their primary sources. The detailed time series of ROx concentrations and chemical budget are summarized in Fig. S3. Figure 6a shows the diurnal variations of the simulated OH, HO$_2$, and RO$_2$. The maximum daily values of OH, HO$_2$, and RO$_2$ concentrations were $2.4\times10^7$, $7.9\times10^8$ and $4.7\times10^8$ molecules·cm$^{-3}$, with the daytime average concentrations of $7.4\times10^6$, $2.4\times10^8$ and $1.7\times10^8$ molecules·cm$^{-3}$, respectively. Model-predicted concentrations of OH in Xiamen were higher than that in the Yellow River Delta (an oil field with high VOCs emission), while the concentrations of HO$_2$ and RO$_2$ showed reverse trends (Chen et al., 2020). The ROx recycling of OH→RO$_2$ was mainly controlled by the reaction of OH+VOCs, and the RO$_2$→HO$_2$ and HO$_2$→OH depended on the reactions with NO (Fig.7). Combined with the ratio of VOCs/NOx (1.1±0.4), it was convinced that NOx would not be the limiting factor in the radical recycling processes. Hence, efficient conversions of radical propagation of RO$_2$+NO→HO$_2$ and HO$_2$+NO→OH were expected, and OH+VOCs→RO$_2$ reaction was the rate-depended step of the radical recycling in our study. The detailed radical chemistry would be further discussed as follows.

Figure 6b shows the daytime average diurnal variations of primary OH sources. HONO photolysis

reached the maximum of 2.7 ppb $h^{-1}$ at around 8:00 LT, which occupied 56±19% of the total OH primary production rates. The second source of OH primary production was $O_3$ photolysis (42±21%), and the percentages of $O_3$+VOCs and OVOCs photolysis were minor. The highest HONO photolysis rate appeared in the morning rush hour, suggesting the influence of vehicle emissions and nocturnal accumulation of HONO (Hu et al., 2020). Considering the radical recycling, the reaction of $HO_2$+NO (8.0±6.2 ppb $h^{-1}$) dominated the total production of OH (Fig. S3a). Meanwhile, OH-initiated oxidations of VOCs (4.9±3.3 ppb $h^{-1}$) consumed OH most during the daytime, followed by OH+CO (2.6±1.9 ppb $h^{-1}$), OH+$NO_2$ (2.4±1.1 ppb $h^{-1}$), OH+NO (0.6±0.3 ppb $h^{-1}$), and OH+$O_3$ (0.2±0.1 ppb $h^{-1}$).

In this study, HCHO photolysis was identified as the most important source for $HO_2$ primary formation, with an average production rate of 1.1±0.6 ppb $h^{-1}$ (Fig.6c), followed by the other OVOCs photolysis (0.4±0.2 ppb $h^{-1}$). The rate of OVOCs photolysis in Xiamen was much lower than that in some megacities, such as Beijing (Liu et al., 2012) and Hong Kong (Xue et al., 2016). The reaction of OH+CO (2.6±2.2 ppb $h^{-1}$) and $RO_2$+NO (2.5±1.5 ppb $h^{-1}$) were also important sources of $HO_2$ (Fig. S3b). The main sink of $HO_2$ was $HO_2$+NO (7.9±6.2 ppb $h^{-1}$), while the loss rates of $HO_2$+$HO_2$ and $HO_2$+$RO_2$ were negligible.

In Fig. 6d, OVOCs photolysis contributed most to primary $RO_2$ production with a rate of 0.5±0.2 ppb $h^{-1}$, accounting for 85±20% of total $RO_2$ primary production. The reaction of unsaturated VOCs and $NO_3$ was the second important source, accounting for 11±18 % of the total primary $RO_2$. The radical recycling rate of OH+VOCs was 8.4 times higher than the sum of $RO_2$ primary production. The consumption reaction of $RO_2$ was mainly caused by $RO_2$+NO (3.7±2.9 ppb $h^{-1}$), and the cross-reactions by ROx themselves were limited.

The daytime average ROx budget and its recycling were also demonstrated (Fig. 7). For the ROx primary sources, the photolysis of HONO (33±14%), $O_3$ (25±13%), HCHO (20±5%) and other OVOCs (17±2%) were the major contributors. For ROx recycling, CO and VOCs reacted with OH producing $HO_2$ and $RO_2$ with the average rates of 4.0 and 4.4 ppbv $h^{-1}$, respectively. $RO_2$+NO and $HO_2$+NO enhanced the production of RO (3.6 ppbv $h^{-1}$) and OH (7.9 ppbv $h^{-1}$), with $O_3$ formed as a by-product. For the termination processes, the reaction rates of ROx and NOx were approximately 2-5 times faster than the cross-reaction rates of ROx.

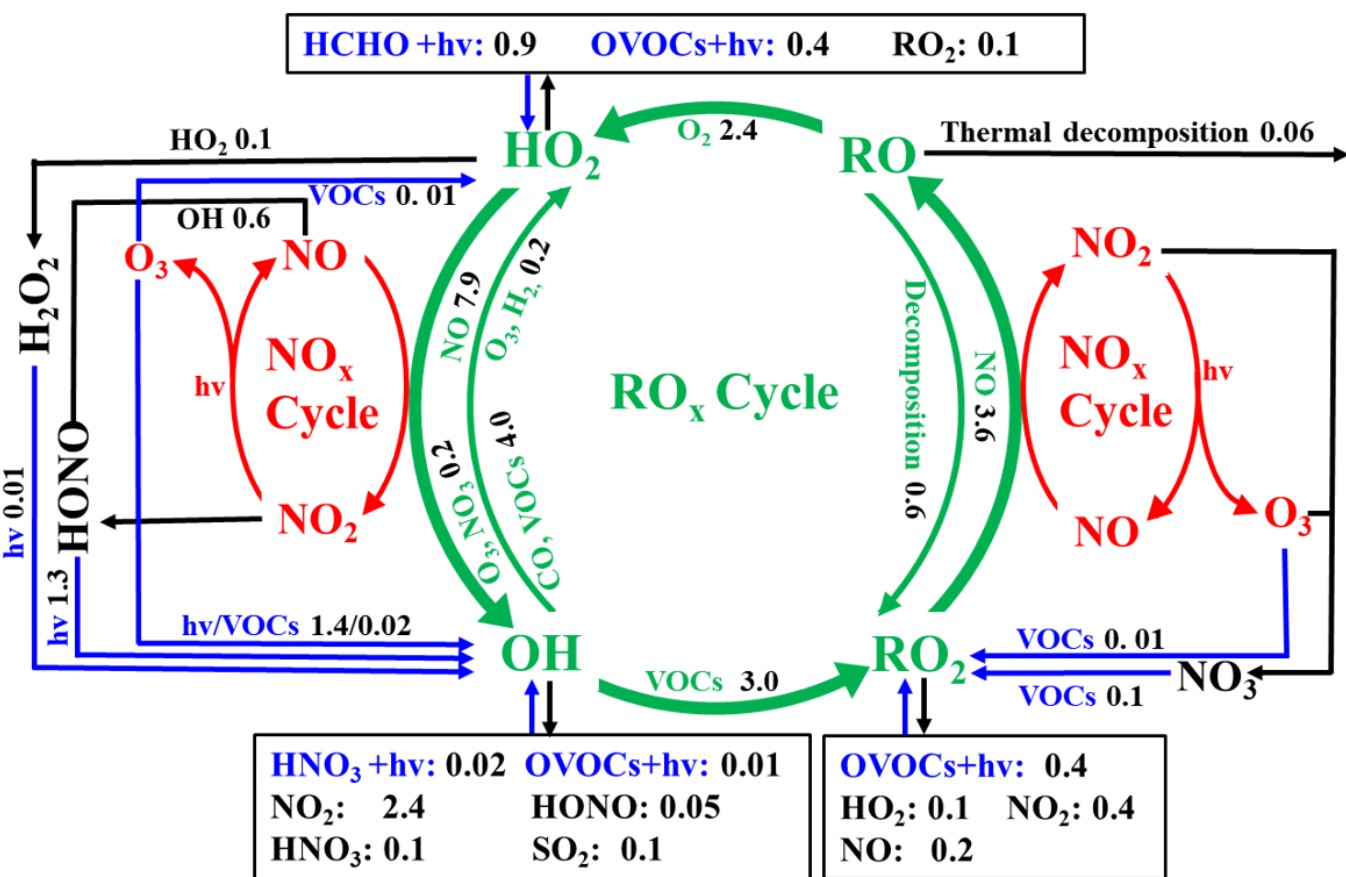


**Figure 7. Daytime ROx budget during 20-29 Sep. 2019 in Xiamen. The unit is parts per billion per hour. The blue,**
**black, and green lines and words indicate the production, destruction, recycling pathways of radicals, respectively.**
## 3.3 $O_3$ formation mechanism
### 3.3.1 Chemical budget and sensitivity analysis of $O_3$ production

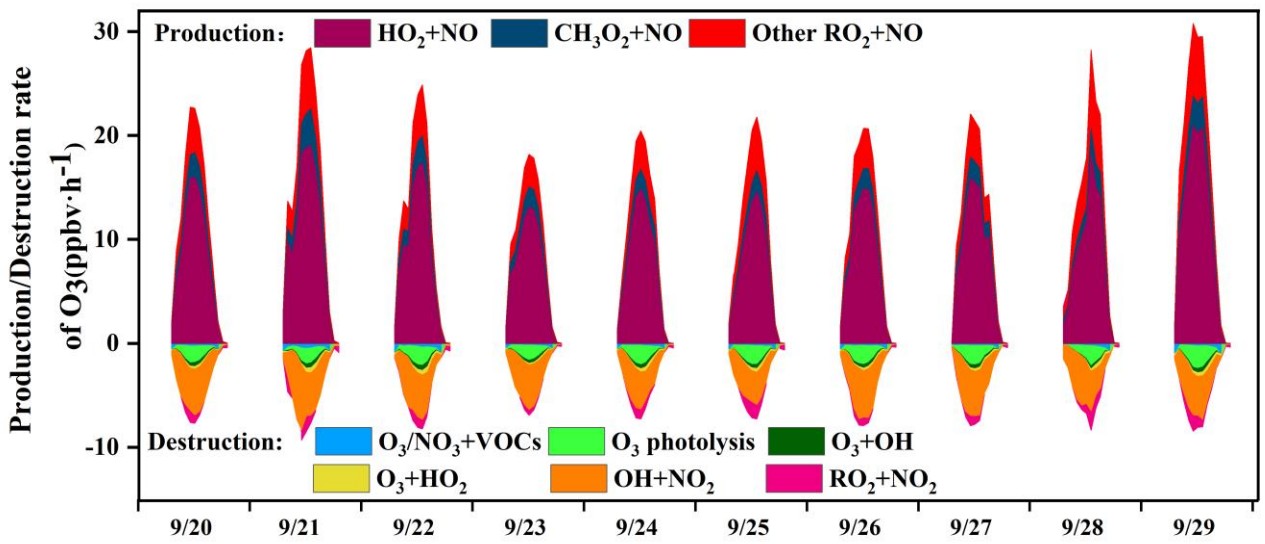


**Figure 8. Time series of model-simulated $O_3$ chemical budgets during 20-29 Sep. 2019 in Xiamen.**
The in situ $O_3$ production mechanism was examined, and the detailed reaction weights were shown

in Fig. 8. The daytime rate of HO$_2$+NO was 7.9±6.2 ppb h$^{-1}$, accounting for 68±4% of the total O$_3$ production. This result was consistent with that in section 3.2.2. The OH radical was the initiator of O$_3$ photochemical formation, and the source of OH from HO$_2$+NO was also the dominant pathway to produce O$_3$ (Liu et al., 2020c). The second pathway of O$_3$ production was RO$_2$+NO (3.6±2.0 ppb h$^{-1}$). The reaction of RO$_2$+NO contained more than 1000 types of RO$_2$ radicals, and the pathway of CH$_3$O$_2$+NO (34±6%) contributed mostly among them. In contrast, the contributors of O$_3$ destruction were OH+NO$_2$ (61±18%), followed by O$_3$ photolysis (18±9%), RO$_2$+NO$_2$ (9±10%), O$_3$+HO$_2$ (4±4%), and O$_3$+OH (4±2%), while the other pathways of O$_3$+VOCs as well as NO$_3$+VOCs contributed limitedly. In addition, the net O$_3$ production (9.1±5.7 ppb h$^{-1}$) in Xiamen was ~2-5 times lower than that derived from the metropolis of Shanghai (26 ppb h$^{-1}$), Lanzhou (23 ppb h$^{-1}$) and Guangzhou (50 ppb h$^{-1}$), reflecting the influence of O$_3$ precursor emissions and photochemical conditions (Xue et al., 2014).

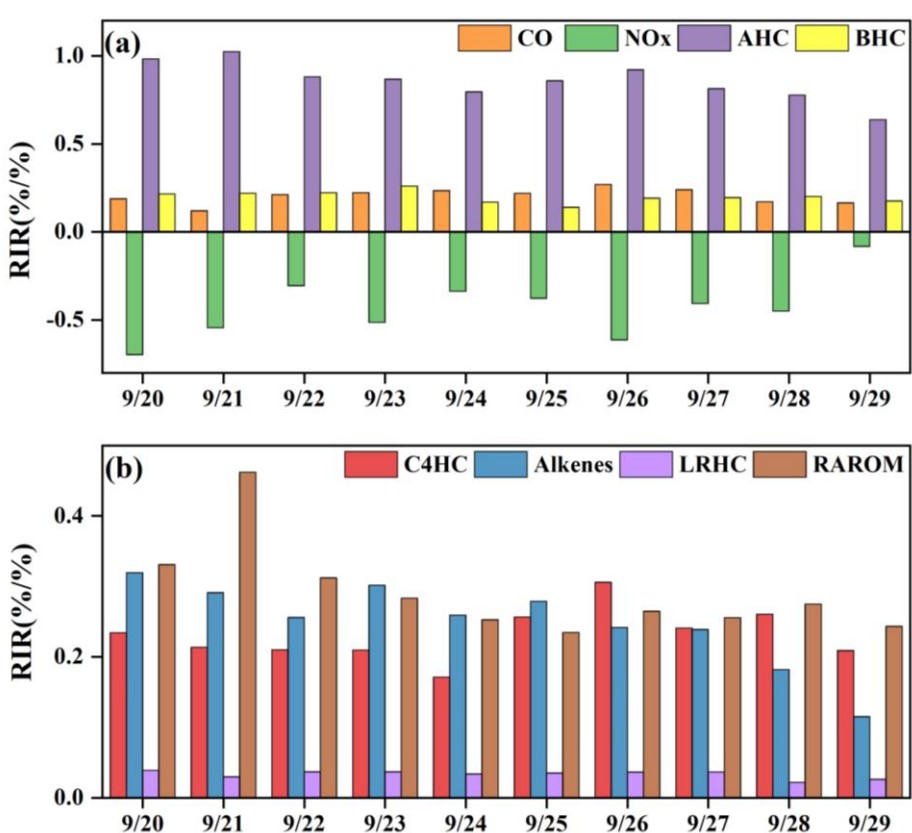

**Figure 9. The model-calculated RIRs for (a) major O$_3$ precursor groups and (b) the AHC sub-groups during high O$_3$ daytime (06:00-18:00 LT) (AHC: anthropogenic hydrocarbons; BHC: biogenic hydrocarbons; RAROM: aromatics except for benzene; LRHC: low reactivity hydrocarbons; C4HC: alkenes, and alkanes with ≥4 carbons).**

In this study, we also calculated the relative incremental reactivity (RIR) to diagnose the sensitivity of O$_3$ formation to its precursors. Figure 9 shows the RIR values for major groups of O$_3$ precursors. Around 50 types of VOCs were classified as anthropogenic hydrocarbons (AHC), and the isoprene was categorized into biogenic hydrocarbons (BHC). Moreover, AHC further divided into four groups of

reactive aromatics (RAROM, including aromatics except for benzene), low reactivity hydrocarbons
(LRHC, including ethane, acetylene, propane, and benzene), alkenes, and alkanes with ≥4 carbons
(C4HC). The in situ $O_3$ production was highly VOCs-sensitive, especially for AHC-sensitive
(0.63−1.02 %/%) (Fig. 9a), followed by CO (0.17−0.27 %/%) and BHC (0.14−0.26 %/%), indicating the
impacts from anthropogenic activities and flourishing vegetation emissions (Liu et al., 2020a; Lin et al.,
2020). The RIRs were NOx-negative ranging from -0.70 to -0.08. As shown in Fig. 9b, the contributors
of AHC sub-groups to RIRs were RAROM (0.24−0.46 %/%), C4HC (0.17−0.30 %/%), alkenes
(0.11−0.32 %/%), and LRHC (0.03−0.04 %/%). Therefore, the reduction of aromatics, alkenes, and
alkanes with ≥4 carbons effectively decreased $O_3$ production, and the reduction of NOx might aggravate
$O_3$ pollution.

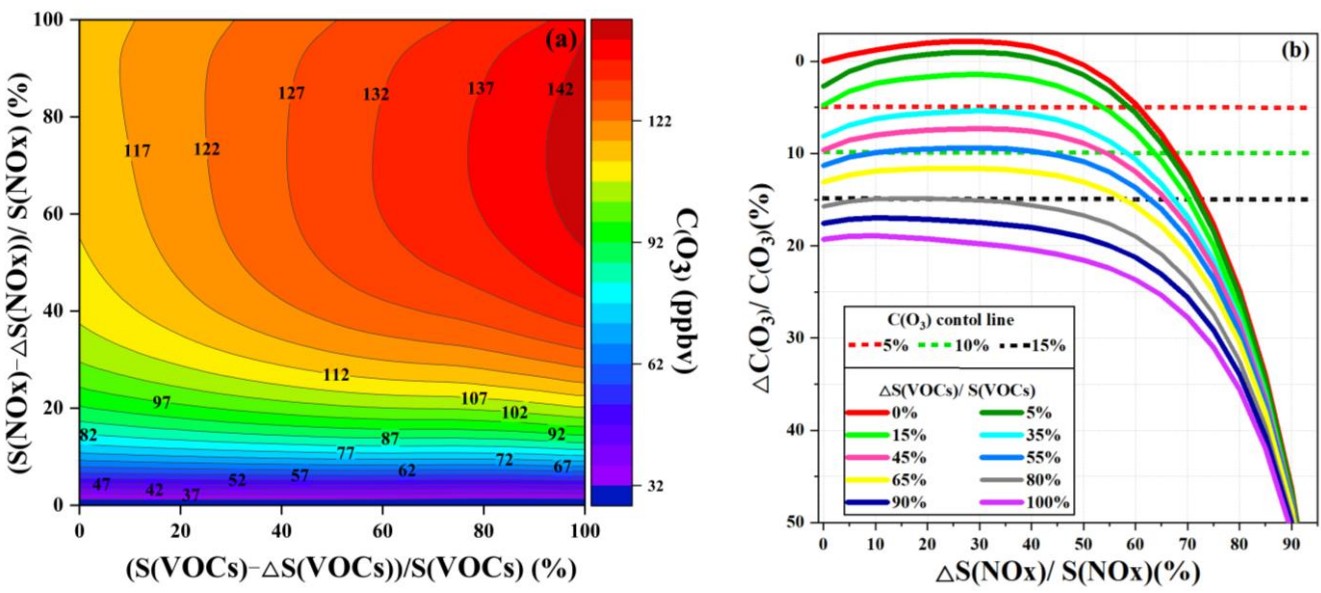

**Figure 10. (a) Isopleth diagrams of modeled $O_3$ production potential ($C(O_3)$) on S(VOCs) and S(NOx) remaining**
**percentages (i.e., (S(VOCs)-S(VOCs))/(S(VOCs)) and (S (NOx)-S(NOx))/(S(NOx)); (b) Relationship of $C(O_3)$**
**increment percentage ($\triangle C(O_3)/C(O_3)$) with S(NOx) and S(VOCs) reduction percentages ($\triangle$S(NOx)/S(NOx) and**
**$\triangle$S(VOCs)/S(VOCs)). Note: $C(O_3)$, S(NOx), and S(NOx) represent the concentrations of corresponding**
**pollutants.**
In order to investigate the $O_3$ control strategies during this multi-day $O_3$ pollution event, the scenario
analysis with reduction by 0-100% at intervals of 5% for the reduction of anthropogenic VOCs
($\triangle$S(VOCs)/S(VOCs) and NOx ($\triangle$S(NOx)/S(NOx)) were conducted using the OBM-MCM. According
to the Empirical Kinetic Modeling Approach (EKMA) and scenario analysis, $O_3$ formation was in the
NO-titration regime (Fig. 10), in accordance with those of RIR analysis, which meant VOCs should be
reduced to effectively control $O_3$ during the $O_3$ pollution event. The maximum value of MDA8h $O_3$ during
the monitoring period was 85 ppbv, exceeding the national air quality standard of 75 ppbv for $O_3$ by 13%.
Hence, the $O_3$ reductions of 5%, 10%, and 15% were set to discuss the reduction schemes of
anthropogenic VOCs and NOx. As shown in Fig. 10b, achieving the 5% control target were 1) S(VOCs)
is reduced by 15%, while S(NOx) remains unchanged; 2) S(VOCs) is reduced larger than 35%; 3) S(NOx)
reduction is higher than 60%. The first scenario of just reducing VOCs emission was the most cost-
efficient way for short-term or emergency control of $O_3$. However, NOx, as important precursors of $PM_{2.5}$,
need to be reduced according to the long-term multi-pollutant control air quality improvement plan in
China, thus the second scenario is a more practical and reasonable way to control air pollution. The 10%
of $O_3$ control target was achieved by the 45% reduction of S(VOCs), and the S(NOx) keeps original
emission. In view of the long-term control strategy of NOx and VOCs, S(VOCs) reduced by 55% and 80%
could decrease 10% and 15% $O_3$ concentrations, respectively. Although VOCs and NOx control measures
were drastically implemented, it is still challenging to achieve the 15% $O_3$ control goals in urban areas
with relatively low precursor emissions. As the episode is a typical pollution process in the coastal region,
the research results might act as reference for the policy makers. Meanwhile, as the $O_3$ sensitivity changed
under the implementation of control measures, it is necessary to adjust timely the reduction of VOC and
NOx policies.
**3.3.2 $O_3$ from local photochemical production and regional transport**

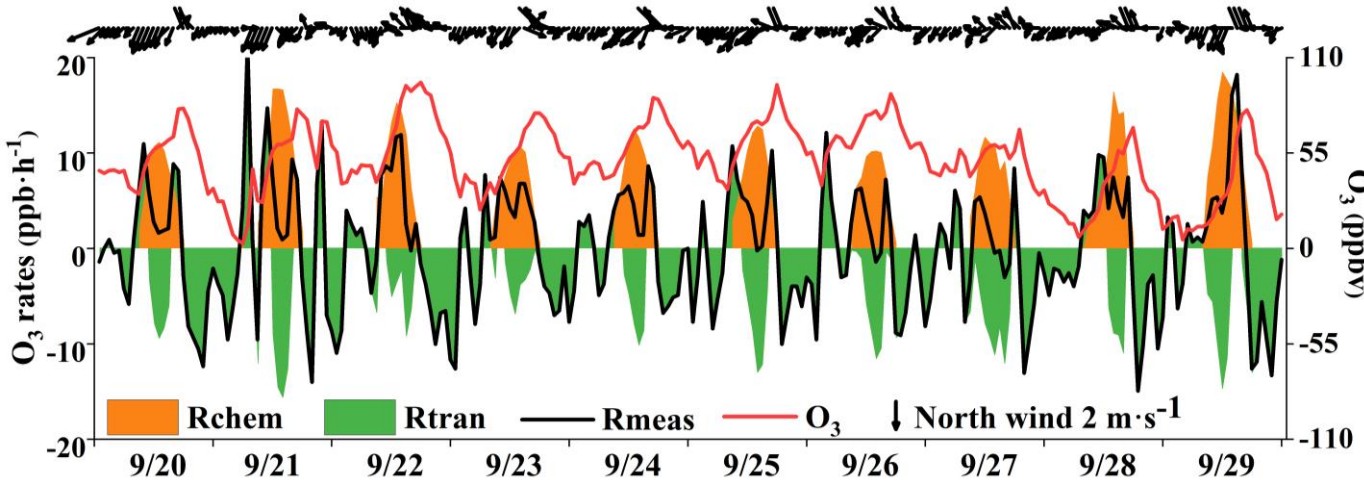


**Figure 11. $O_3$ accumulation and contributions from local photochemical production and regional transport, and**
**Rchem, Rtran, and Rmeas in figure caption represent local $O_3$ photochemical production, regional transport and**
**observed $O_3$ formation rate, respectively.**
Previous studies have found that the variation of $O_3$ mixing ratios was mainly influenced by chemical
and physical processes (Xue et al., 2014; Tan et al., 2018). Figure 11 shows the time series of $O_3$
accumulation and contributions from local photochemical production and regional transport. The
observed rate of change in $O_3$ (Rmeas) was calculated by the derivative of the observed $O_3$ concentrations
(Rmeas=d($O_3$)/dt). The local $O_3$ production (Rchem) was calculated by Equation 3, and computed hourly
by the OBM as described in Section 2.2. The physical processes (Rtran) were calculated by the equation

of Rtran=Rmeas−Rchem, including horizontal and/or vertical transport, dry deposition dilution mixing, and so on. Many studies showed that the impacts of dry deposition were minor, thus the differences between observed $O_3$ changes and local $O_3$ production were mainly caused by the regional transport (note that the effect of atmospheric mixing was also included in this term), which could be treated as regional transport and could reasonably quantify the contributions of regional transport at our observation site (Zhang et al., 2021; Chen et al., 2020). The positive values of Rtran represented the $O_3$ import of regional transport, while the negative values indicated the $O_3$ export and deposition. We quantified the contributions of local photochemical formation and regional transport to the observed $O_3$, and figured out the reasons for the $O_3$ pollution process.

As shown in Fig. 11, two regular $O_3$ import phenomenon with positive values of Rtran were observed, and the curve of the Rmeas showed the "M" trend during the daytime. The first transient intense $O_3$ import happened in the early morning (at around 6:00-9:00), leading to a rapid increase in $O_3$ concentration, which was mainly attributed to the residual ozone from the day before. The $O_3$ export was remarkable at around 10:00-16:00, indicating the potential impacts on air quality in downwind areas. Generally, the maximum daily value of $O_3$ at this observation site appeared at around 15:00 LT without regional transport (Wu et al., 2019). In Figure 11, we found that the $O_3$ concentrations showed two peaks at around 15:00 and 17:00 LT, and $O_3$ concentrations rose slowly, or even decreased firstly and then increased between the two peaks. Under these circumstances, the local photochemical production kept producing $O_3$, but the decreased $O_3$ concentrations could be attributed to the favorable atmospheric conditions in diluting pollutants ($O_3$ export). When the near-surface wind direction changed from northeast to southeast, the second $O_3$ import phenomenon occurred in the afternoon (16:00-19:00 LT) in four days (20, 25 27 and 29 Sep.). Due to the persistence of Rtran in the afternoon, the daily maximum $O_3$ values appeared at around 17:00 LT. Under the conditions of southeast wind direction, downtown area with high density vehicles would make $O_3$ and its precursors transmitting to our observation site, consistent with the diurnal patterns of $NO_2$, OVOCs, alkanes, and aromatic in the early morning and afternoon (Fig. 2) to match with the "M" trend of Rmeas. This result indicated that the sudden changes of near-surface winds were corresponding to the variation in the transport of the urban plume.

According to the synoptic situations and meteorological parameters (Fig. 1, Fig. S4 and Fig.12), the environmental conditions also favored the $O_3$ pollution process during the observation periods. The contribution of Rchem (daily maximum: ranged from 10.2 to 19.1 ppb h$^{-1}$) during the daytime was observed (Fig.11). In Fig. S4abc, the monitoring site was continuously affected by the northerly airflow

with high $O_3$ and its precursors (from an industrial city adjacent to Xiamen of Quanzhou or polluted
regions of Yangtze River Delta), due to the typhoon 'Tapah' from 20 to 22 Sep. 2019. The transport of $O_3$
import appeared on 21 Sep. ($7.1\pm7.0$ ppb $h^{-1}$), which resulted in the accumulation of $O_3$ (the MDA8h $O_3$:
85 ppbv) on 22 Sep. When the influence of typhoon disappeared, the direction of airflow turned from
northerly into southwest with humid and warm at 500hPa (Fig. S4d), the surface wind on Sep. 23 was
affected by the control of the cold northerly airflow (Fig. S4ef). Meteorological conditions including
continental high pressure during 23 to 27 Sep. were favorable to the accumulation of air pollutants (Fig.
12). The isoline of 5880 gpm moving from north to the Yangtze River (Fig. 12a,b) indicated the
strengthened subtropical high pressure during 23-27 Sep. 2019, which carried high temperature, low RH,
and stagnant weather conditions, and the transport rate of $O_3$ export ($5.4\pm3.4$ ppb $h^{-1}$) on 24-26 Sep. was
lower than that on other days ($6.3\pm4.0$ ppb $h^{-1}$). Favorable meteorological conditions significantly
affected the formation and accumulation of $O_3$, and we chose five meteorological parameters (i.e. UV, T,
RH, P and WS) to quantify the complex nonlinear relationships between $O_3$ and its influencing factors
based on a generalized additive model (GAM) (Hua et al., 2021). Table S3 showed that the factors had
significant non-linear impacts on $O_3$ concentration changes at the level of P-value<0.01 and degrees of
freedom>1, indicating that each influencing factor has statistical significance as an explanatory variable.
According to the F-values reflecting the importance of the influencing factors, the orders of the
explanatory variables were RH (40.1) > WS (26.9) > T (10.9) > P (3.9) > UV (3.0). Response curves of
$O_3$ concentration to explanatory factors are presented in Fig. 13. The $O_3$ concentration showed a
remarkable upward trend until the UV increased to 17 W·m$^{-2}$, then changed little with the fluctuation of
UV (Fig. 13a). In previous studies, UV had a significant positive correlation with $O_3$ concentrations (Ma
et al., 2020), and these results showed the regional transport impacts on $O_3$ formation in our study. The
RH and T had negative and positive correlations with $O_3$ concentrations, respectively (Fig. 13b and Fig.
13c). The increase of wind speed was favorable for $O_3$ regional transport (Fig. 13d). The influence of
atmospheric pressure on $O_3$ seemed to be irregular and minor, which could be ignored (Fig. 13e). Hence,
under the combined effects of favorable photochemical reaction conditions and strengthened WPSH, the
MDA8h $O_3$ exceeded the standard of 75 ppbv during 24-26 Sep. Previous studies had found that severe
multi-day $O_3$ pollution appeared under the WPSH control (Wang et al., 2018a). Overall, the results
indicate that the three conditions of local photochemical production, synoptic situations and regional
transport played very important roles in the pollution event.

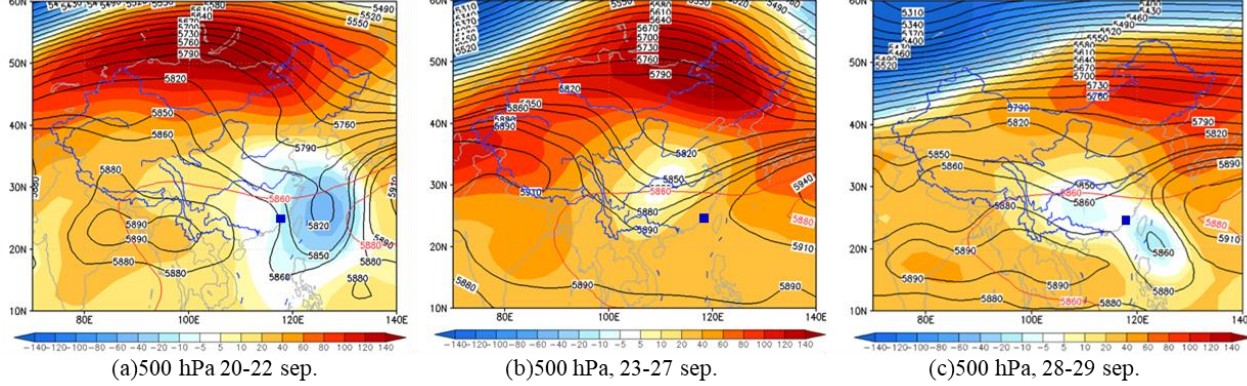

**Figure 12. Synoptic situations of continental high pressure from 20 to 29 Sep. 2019. The gradient color area indicates the WPSH over the map and the contour line was from the characteristic isoline of 5880 gpm to the center isoline of 5920 gpm. The blue square is the study site.**

**Figure 13. Response curves in GAM model of O$_3$ concentration to changes in (a) ultraviolet radiation (UV), (b) relative humidity (RH), (c) temperature (T), (d) wind speed (WS), and (e) pressure (P). The y-axis is the smoothing function values. The x-axis is the influencing factor; the vertical short lines represent the concentration distribution characteristics of the explanatory variables; the shaded area around the solid line indicates the 95%**

**confidence interval of $O_3$ concentration.**

## 4 Conclusions

In the present study, we analyzed a typical high $O_3$ event during 20-29 Sep. 2019 in a coastal city of Southeast China. We clarified the characteristics of AOC, OH reactivity, and radical chemistry, as well as $O_3$ formation mechanisms using the OBM-MCM model. The predominant oxidant for AOC during the daytime and nighttime was the OH and $NO_3$, respectively. During the period of $O_3$ pollution process, OVOCs, $NO_2$, and CO consumed OH mostly. Meanwhile, the photolysis of HONO, $O_3$, HCHO, and other OVOCs were major sources of ROx, which played the initiation roles in atmospheric oxidation processes. The radical termination reactions were governed by cross-reactions between ROx and NOx. The RIRs and EKMA results showed that the $O_3$ formation in autumn in the coastal city was VOCs-sensitive, and the VOCs were the limited factor of radical recycling and $O_3$ formation. The reduced emissions of aromatics, alkenes, and alkanes with ≥4 carbons were benefit for ozone pollution control. The three conditions of local photochemical production, synoptic situations and regional transport played very important roles in the pollution event. Overall, the results clarified the $O_3$ pollution process with relatively low local precursor emissions, and implied the fact that $O_3$ pollution control in coastal cities needs to be further studied.

## Code/Data availability

The observation data at this site are available from the authors upon request.

## Authorship Contribution Statement

Taotao Liu and Youwei Hong contributed equally to this work. Jinsheng Chen and Likun Xue designed and revised the manuscript. Taotao Liu collected the data, contributed to the data analysis. Taotao Liu and Youwei Hong performed chemical modeling analyses of OBM-MCM and wrote the paper. Jinsheng Chen supported funding of observation and research. Lingling Xu, Mengren Li, Chen Yang, Yangbin Dan, Yingnan Zhang, and Min Zhao contributed to discussions of results. Zhi Huang and Hong Wang provided meteorological conditions in Xiamen.

## Competing interests

The authors declare that they have no conflict of interest.

**Acknowledgments**

This study was funded by the Cultivating Project of Strategic Priority Research Program of Chinese Academy of Sciences (XDPB1903), the FJIRSM&IUE Joint Research Fund (RHZX-2019-006), the Center for Excellence in Regional Atmospheric Environment, CAS (E0L1B20201), the Xiamen Youth Innovation Fund Project (3502Z20206094), the foreign cooperation project of Fujian Province (2020I0038) and Xiamen Atmospheric Environment Observation and Research Station of Fujian Province.

**Supplementary information**

Attached please find supplementary information associated with this article.

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
