# Peer review of "Atmospheric oxidation capacity and ozone pollution mechanism in a"

_Atmospheric Chemistry and Physics, 2021_

## Author Comment (AC1)

**Response to Reviewers**

Comment on acp-2021-764

**Anonymous Referee #2**

Liu et al. presented a typical ozone pollution event study of a coastal city of southeast China for the exploration of AOC, OH reactivity, radical chemistry and ozone pollution mechanism with OBM-MCM method. The predominant oxidant for AOC, dominant contributor for OH reactivity, important source of ROx radical were examined, as well as the ozone formation regime sensitivity. Finally, the VOCs emission reduction were proposed for limiting the radical recycling and $O_3$ formation. Overall, the paper is appropriate for publication at ACP subject to the following concerns.

**Response:** Thank you very much for your exploratory and constructive advice. Here, we have carefully revised the manuscript.

**Specific comments:**

Even though this paper clarifies several important characteristics and mechanism of the ozone pollution for a selected case, the representativeness for a short period and the specific location seems not to be abroad of interests. I would like to suggest the authors can enhance the significance of the findings for the readership.

**Response:** We thank the reviewer for the comments which are helpful for us to improve the paper. We have further revised the manuscript accordingly, and hope meet with approval.

Regarding to the location, the authors considered the site shows a relatively low $O_3$ precursors and complex meteorological conditions. However, no evidence was found for the comparison of levels of $O_3$ precursors, and also the impacts of complex meteorological conditions were not well discussed. These may be improved via, e.g.: (1) comparative study on the non-low levels of $O_3$ precursors case for the ozone pollution; (2) the impacts of change of meteorological conditions (not only the synoptic situation) on the ozone pollution.

**Response:** Thanks for your suggestion. Based on your suggestions, we made the following changes.

(1)
The comparison of NO, $NO_2$ and total VOCs levels in cities between China and other countries is listed in Table 1. The comparison indicated relatively low $O_3$ precursor emissions in our observation site. And the detailed comparative discussion on the nonlow levels of O$_3$ precursors case for the ozone pollution was also added to the revised manuscript.

"In a coastal city of Southeast China, the concentrations of O$_3$ precursors were higher than those in remote sites and background, but lower than those in most of urban and suburban areas, even lower than those in rural regions (Table S1). In a word, O$_3$ precursor emissions in our observation site were relatively low."

"The concentration of TVOCs in Xiamen (17.2±4.8 ppbv) was lower than that in the developed areas with large anthropogenic emissions (i.e., Beijing (41.2 ppbv), Lanzhou (45.3 ppbv), Wuhan (30.2 ppbv), Chengdu (36.0 ppbv), Hong Kong (26.9 ppbv), Los Angeles (41.3 ppbv) and Tokyo (43.4 ppbv), comparable to some urban with low pollution emissions (i.e., Wuhan (30.2 ppbv), Chengdu (36.0 ppbv), Hong Kong (26.9 ppbv), Los Angeles (41.3 ppbv) and Tokyo (43.4 ppbv)), but was higher than that at the background and remote sites (i.e., Mt. Wuyi (6.1 ppbv) and Mt. Waliguan (2.6 ppbv)) (Table S1)."

Table S1 Comparison of NO, NO$_2$ and total VOCs levels in cities between China and other countries (Unit: ppbv).

| Location | NO$_2$ | NO | VOCs | Site category | Observation periods | Reference |
|---|---|---|---|---|---|---|
| Xiamen | 15.4 | 1.4 | 17.2 | Urban | Sep. 2019 (episode) | This study |
| Beijing | 16.8 | 2.1 | 44.2 | Urban | | Liu et al., 2021b |
| Wuhan | 17.5 | 3.2 | 30.2 | Urban | Summer 2018 (episode) | Liu et al., 2021b |
| Lanzhou | 15.8 | 2.9 | 45.3 | Urban | | Liu et al., 2021b |
| Shanghai | 14.2 | 3.38 | 25.3 | Urban | Jun. 2019 (episode) | Zhu et al., 2020 |
| Chengdu | 39.0 | 3.6 | 36.0 | Urban | Jul. 2017 (episode) | Yang et al., 2020 |
| Los Angeles | - | - | 41.3 | Urban | May. to Jun. 2010 | Warneke et al., 2012 |
| London | - | - | 22.1 | Urban | 1998–2008 | Von Schneidemesser et al., 2010 |
| Tokyo | - | - | 43.4 | Urban | 2003–2005 | Hoshi et al., 2008 |
| Beijing | 11.5 | 4.8 | 28.1 | Suburban | Aug. 2018 | Yang et al., 2021 |
| Hong Kong | 25.0 | 14.0 | 26.9 | Suburban | Aug. to Nov. 2013 | Wang et al., 2018 |
| Chengdu | 11.4 | 8.0 | 28.0 | Suburban | Summer 2019 | Yang et al., 2021a |
| Qingdao | 16.7 | 1.6 | 7.6 | Rural | Oct. to Nov. 2019 | Liu et al., 2021a |
| The Pearl River Delta | 39.9 | 4.2 | 38.0 | Rural | Octo. to Nov. 2014 | He et al., 2019 |
| Hong Kong | 12.2 | 1.9 | 10.9 | Regional background | Aug. to Dec. 2012 | Li et al., 2018 |
| Mt. Wuyi | - | - | 4.7 | Background | Dec. 2016 | Hong et al., 2019 |
| Mt. Tai | - | - | 8.8 | Background | Jun. 2006 | Suthawaree et al., 2010 |
| Mt. Waliguan | - | - | 2.6 | Remote region | Jul. to Aug. 2003 | Xue et al., 2013 |

Note: "-" means that the data was not mentioned in the relevant studies.

(2)
We strongly agree with your suggestions of strengthening the analysis of meteorological conditions. During the observation periods, Xiamen was affected by

various meteorological conditions, such as typhoon and the West Pacific Subtropical High (WPSH) accompanied by temperature inversion phenomenon, thus we focused on the analysis of meteorological conditions and ignored the conventional analysis of other meteorological parameters (wind speed (WS), air temperature (T), pressure (P), relative humidity (RH), and photolysis rate constants). Hence, we used the Generalized Additive Model (GAM) to study the influencing factors on $O_3$ pollution. GAM model has been widely used in $O_3$ pollution research, and can deal with the complex nonlinear relationship between $O_3$ and its influencing factors effectively (Hua et al., 2021; Ma et al., 2020). The detailed discussion was shown in the manuscript of Section 3.3.2, and the main revisions are as follows.

"Favorable meteorological conditions significantly affected the formation and accumulation of $O_3$, and we chose five meteorological parameters (i.e. UV, T, RH, P and WS) to quantify the complex nonlinear relationships between $O_3$ and its influencing factors based on a generalized additive model (GAM) (Hua et al., 2021). Table S3 showed that the factors had significant non-linear impacts on $O_3$ concentration changes at the level of P-value<0.01 and degrees of freedom>1, indicating that each influencing factor has statistical significance as an explanatory variable. According to the F-values reflecting the importance of the influencing factors, the orders of the explanatory variables were RH (40.1) > WS (26.9) > T (10.9) > P (3.9) > UV (3.0). Response curves of $O_3$ concentration to explanatory factors are presented in Fig. 13. The $O_3$ concentration showed a remarkable upward trend until the UV increased to 17 $W·m^{-2}$, then changed little with the fluctuation of UV (Fig. 13a). In previous studies, UV had a significant positive correlation with $O_3$ concentrations (Ma et al., 2020), and these results showed the regional transport impacts on $O_3$ formation in our study. The RH and T had negative and positive correlations with $O_3$ concentrations, respectively (Fig. 13b and Fig. 13c). The increase of wind speed was favorable for $O_3$ regional transport (Fig. 13d). The influence of atmospheric pressure on $O_3$ seemed to be irregular and minor, which could be ignored (Fig. 13e)."

Table S3 Estimated degree of freedom (Edf), degree of reference (Ref. df), P-value, F-value, deviance explained (%), adjusted $R^2$ for the smoothed variables (including UV, T, RH, P, and WS) in the GAM model.

| Smoothed variables | [a]Edf | [a]Ref.df | [b]F | [c]P-value | [d]Adjust $R^2$ | [e]Deviance explained (%) |
|---|---|---|---|---|---|---|
| UV ($W·m^{-2}$) | 3.1 | 3.8 | 3.0 | 0.0 | 0.0 | 5.4 |
| T (°C) | 5.3 | 6.5 | 10.9 | 0.0 | 0.2 | 24.1 |
| RH (%) | 2.9 | 3.6 | 40.1 | 0.0 | 0.4 | 38.9 |
| WS ($m·s^{-1}$) | 2.9 | 3.6 | 26.9 | 0.0 | 0.3 | 29.3 |
| P (hPa) | 6.9 | 8.0 | 3.9 | 0.0 | 0.1 | 13.4 |

Note: [a] The degree of freedom (edf, ref.df) of the explanatory variable is 1, indicating the linear relationships between the explanatory variable and the response variable, and a non-linear relationship is shown when the degree>1; [b] a high F-value indicates the great importance of the influencing factor; [c] the P-value is used to judge the significance of the model result; [d] the adjusted

$R^2$ is the value of the regression square ranging from 0 to 1; [e] the deviance explained represents the fitting effect.

[Figure]

**Figure 13. Response curves in GAM model of O₃ concentration to changes in (a) ultraviolet radiation (UV), (b) relative humidity (RH), (c) temperature (T), (d) wind speed (WS), and (e) pressure (P). The y-axis is the smoothing function values. The x-axis is the influencing factor; the vertical short line represents the amount of data; the shaded area around the solid line indicates the 95% confidence interval of O₃ concentration.**

I could not find the observed HCHO data in the paper, which is very important for the observation constrained modeling, and further discussion on the radical sources and evaluation of the highest OFP species.

**Response:** Thank you for your suggestions. We strongly agree with this idea that HCHO is very important for observation constrained modeling.

The gas chromatography-mass spectrometer (GC-FID/MS, TH-300B, Wuhan, CN)

used for atmospheric VOCs concentrations monitoring cannot detect HCHO in this study. When the HCHO concentrations were not observed, the concentrations could be locally and reasonably calculated by the model according to the other observed pollutants of $O_3$ precursors (Table 2). Some studies exploring the $O_3$ formation mechanism based on OBM model also did not observe HCHO data (Chen et al., 2020, Liu et al., 2021; Li et al., 2018; Wang et al., 2020). Meanwhile, we strongly agree with your idea and realized the importance of HCHO in $O_3$ formation, hence our team improved the monitoring of *Atmospheric Formaldehyde Online Analyzer* and *Chemical Ionization Mass Spectrometry (CIMS)* in May 2021. A more optimized and complete monitoring system is also the future optimization goal of our model.

Meanwhile, the index of agreement (IOA) can be used to judge the reliability of the model simulation results, and its equation is (Liu et al., 2019).

$$IOA = 1 - \frac{\sum_{i=1}^{n}(O_i - S_i)^2}{\sum_{i=1}^{n}(|O_i - \bar{O}| - |S_i - \bar{O}|)^2} \tag{4}$$

where $Si$ is simulated value, $Oi$ represents observed value, $\bar{O}$ is the average observed values, and n is the sample number. The IOA range is 0-1, and the higher the IOA value is, the better agreement between simulated and observed values is. In many studies, when IOA ranges from 0.68 to 0.89 (Wang et al., 2018a), the simulation results are reasonable, and the IOA in our research is 0.80. The hourly simulated and observed $O_3$ during the observation periods at the study site in Figure R1 showed that the performance of the OBM-MCM model was reasonably acceptable.

[Figure]

Figure R1. The hourly simulated and observed $O_3$ during the observation periods at the study site.

The OFP results had relatively great errors brought by the missing data of HCHO. In this study, we only calculated the OFP values briefly and did not analyze them in-depth, which could not help my analysis well and even confuse readers. Anymore, the OH reactivities and RIRs can better reflect the importance of its precursors for $O_3$ formation. Hence, we think it is a better choice to delete the analysis of OFP from the revised manuscript, which can help readers better understand the full text.

OBM modeling: Please specify the setting of dry deposition velocity.

**Response:** Thanks for your suggestion. The specific setting of dry deposition velocity was shown in the supporting information (Table S2).

Table S2. Dry deposition velocity (cm s$^{-1}$) for chemical species (Zhang et al., 2003).

| Symbol | Name | dry deposition velocity |
|---|---|---|
| $O_3$ | Ozone | 0.6 |
| $NO_2$ | Nitrogen dioxide | 0.6 |
| HONO | Nitrous acid | 1.9 |
| $HNO_3$ | Nitric acid | 4.7 |
| $HNO_4$ | Pernitric acid | 3.3 |
| $NH_3$ | Ammonia | 1 |
| $SO_2$ | Sulphur dioxide | 0.8 |
| $H_2SO_4$ | Sulphuric acid | 1.1 |
| $H_2O_2$ | Hydrogen peroxide | 1.2 |
| PAN | Peroxyacetylnitrate | 0.4 |
| PPN | Peroxypropylnitrate | 0.4 |
| APAN | Aromatic acylnitrate | 0.5 |
| MPAN | Peroxymethacrylic nitric anhydride | 0.3 |
| HCHO | Formaldehyde | 0.9 |
| MCHO | Acetaldehyde | 0.2 |
| PALD | C3 Carbonyls | 0.2 |
| C4A | C4-C5 Carbonyls | 0.2 |
| C7A | C6-C8 Carbonyls | 0.2 |
| ACHO | Aromatic carbonyls | 0.2 |
| MVK | Methyl-vinyl-ketone | 0.2 |
| MACR | Methacrolein | 0.2 |
| MGLY | Methylgloxal | 0.2 |
| MOH | Methyl alcohol | 0.7 |
| ETOH | Ethyl alcohol | 0.6 |
| POH | C3 alcohol | 0.5 |
| CRES | Cresol | 0.2 |
| FORM | Formic acid | 1.4 |
| ACAC | Acetic acid | 1.1 |
| ROOH | Organic peroxides | 0.6 |
| ONIT | Organic nitrates | 0.4 |
| INIT | Isoprene nitrate | 0.3 |

Line 47, "&" may be not the suitable format for the text. Btw, here the authors want to indicate the "temporal and spatial distribution" of what? Ozone concentration? or precursors? Please clarify it.

**Response:** As you suggested, we have clarified the temporal and spatial distribution, and the main revisions are as follows.

"$O_3$ formation is affected by multiple factors such as $O_3$ precursor speciation or level, atmospheric oxidation capacity, meteorological conditions and regional transport."

Line 139-148, Please list the relevant reaction and reaction rates in the Eq. 1 to Eq. 3, at least in the Supplementary.

**Response:** Thanks for your suggestion. The relevant reaction and reaction rates were listed in Table 1, and the main revisions are as follows.

"Table 1 shows the production and destruction reactions and relevant reaction rates of $O_3$ in our study. The production rate of $O_3$ ($P(O_3)$) includes $RO_2+NO$ (R1) and $HO_2+NO$ reactions (R2, Eq. 1), and the destruction of $O_3$ ($D(O_3)$) involves reactions of $O_3$ photolysis (R3), $O_3+OH$ (R4), $O_3+HO_2$ (R5), $NO_2+OH$ (R6), $O_3+VOCs$ (R7), and $NO_3+VOCs$ (R8, Eq. 2). The net $O_3$ production rate ($P_{net}(O_3)$) is calculated by $P(O_3)$ minus $D(O_3)$ as equation 3."

**Table 1 Simulated production and destruction reactions and relevant reaction rates of $O_3$.**

| Reactions | Reaction rates | Number |
|---|---|---|
| **$O_3$ production pathways-$P(O_3)$** | | |
| $RO_2+NO \rightarrow RO+NO_2$ | $2.7\times10^{-12}\times EXP(360/T)$ | R1 |
| $HO_2+NO \rightarrow OH+NO_2$ | $3.45\times10^{-12}\times EXP(270/T)$ | R2 |
| **$O_3$ destruction pathways-$D(O_3)$** | | |
| $O_3+hv \rightarrow O^1D+O_2$ | $JO^1D$ | R3a |
| $O^1D+H_2O \rightarrow OH$ | $2.14\times10^{-10}$ | R3b |
| $O_3+OH \rightarrow HO_2$ | $1.70\times10^{-12}\times EXP(-940/T)$ | R4 |
| $O_3+HO_2 \rightarrow OH$ | $2.03\times10^{-16}\times(T/300)^{4.57}\times EXP(693/T)$ | R5 |
| $NO_2+OH \rightarrow HNO_3$ | $3.2\times10^{-30}\times9.7\times10^{18}\times P/T\times(T/300)^{-4.5}\times3.0^{-11}\times10^{\log10(0.41)}/(1+(\log(3.2^{-30}\times9.7E\times10^{18}\times P/T\times(T/300)^{-4.5}\times3.0^{-11}/(0.75-1.27\times(\log_{10}(0.14))^2)/(3.2^{-30}\times9.7E\times10^{18}\times P/T\times(T/300)^{-4.5}+3.0^{-11})$ | R6 |
| $O_3+VOCs \rightarrow Carbonyls+Criegee\ biradical$ | $Kcons.1$ | R7 |
| $NO_3+VOCs \rightarrow RO_2$ | $Kcons.2$ | R8 |

Note: The reaction rates of Kcons.1 and Kcons.2 were constant. There were around 700 reactions of $VOCs+NO_3/O_3$, and the relevant reaction rates were different constants, which can be obtained from this website http://mcm.leeds.ac.uk/MCM/.

Line 234-239, High AOC were calculated for the ozone pollution episode in this study, which is significantly higher than those at Hongkong, Shanghai, etc. However, as stated in the introduction, the AOC levels in the polluted regions are much higher than those

at the background sites or remote regions. Does it mean that this site can be classified as a polluted one? And contradict to that non-low level of precursors? The authors should discuss carefully what are the main reasons causing the high AOC in this study.

**Response:** Thanks for your suggestion. We apologize for the confusion caused by the incorrect AOC calculation and inappropriate comparison of AOC among different cities in my study.

The AOC is calculated as the sum of oxidation rates of various primary pollutants (CO, $NO_x$, VOCs, etc.) by the major oxidants (i.e., OH, $O_3$, $NO_3$), which did not list the types of VOCs in detail. In fact, the species of VOCs in AOC calculation mainly include alkanes, alkenes, aromatics and OVOCs, while we computed AOC using many VOCs that should not be considered in AOC calculation, so that the AOC levels in our study were overestimated. We recalculated AOC (Fig. 3) and have corrected it in the manuscript. After comparison of the recalculated AOC, the concentrations of $O_3$ precursors in Xiamen were lower than those in Hong Kong and Shanghai we mentioned, but the AOC levels in our study were comparable to or even lower compared with the AOC in Hong Kong and Shanghai. According to the AOC definition, the key factors to quantify AOC are processes and rates of species being oxidized in the atmosphere (Liu et al., 2021c). Hence, the factors of photolysis rate, meteorological conditions, pollutant concentrations and regional transport would influence the AOC levels, and we cannot think the high AOC value means the polluted levels of the regions. When we compare the AOC among different sites, we should compare the daily maximum AOC and also analyze other relevant information, such as site category, solar radiation, pollutant concentrations. As Table R1 shown, although the levels of $O_3$ precursors in these urban sites were higher than those in Xiamen, the photolysis rates in these cities were lower than those in Xiamen. The detailed discussions were shown in the manuscript.

Table R1 Comparison of NO, $NO_2$, total VOCs (ppbv), AOC (molecules $cm^{-3}$ $s^{-1}$) and $J(NO_2)$ levels in Xiamen, Shanghai and Hong Kong.

| Location | $NO_2$ | NO | VOCs | Site category | AOC | Maximum AOC | $J(NO_2)$ ($10^{-3}$ $s^{-1}$) | Maximum $J(NO_2)$ | Reference |
|---|---|---|---|---|---|---|---|---|---|
| Xiamen | 15.4 | 1.4 | 17.2 | Urban | $6.7\times10^7$ | $1.3\times10^8$ | 3.5 | 11.1 | This study |
| Shanghai | 14.2 | 3.4 | 25.3 | Urban | $3.9\times10^7$ | $1.0\times10^8$ | 2.8 | 8.0 | Liu et al., 2020 |
| Hong Kong | - | - | 32.7 | Urban | $6.3\times10^7$ | $1.3\times10^8$ | - | 6.0 | Xue et al., 2016 |
| Hong Kong | 12.2 | 1.9 | 10.9 | Regional background | $1.6\times10^7$ | $6.2\times10^7$ | 2.3 | 9.3 | Li et al., 2018 |

Note: "-" means that the data was not mentioned in the relevant studies.

[revised manuscript text omitted]

Fig. 11, The Rtran was determined by the difference of Rmeas and Rchem. So my main concern is that how about the accuracy of Rtran? At least, I think it include the considerable uncertainties of Rchem. The inference about transport amount need be

more cautious. Also no evidence provided can prove the northerly air flow is ozone polluted. Secondly, the authors explained why the two regular $O_3$ important phenomenon with positive Rtran happened. However, why did negative Rtran observed around noontime every day?

**Response:** Thanks for your suggestion. We strongly agree that there were uncertainties in the model simulation.

Firstly, the observation data of the gaseous pollutants (i.e., $O_3$, CO, NO, $NO_2$, HONO, $SO_2$, and VOCs), meteorological parameters (i.e., T, P, and RH), and photolysis rate constants ($J(O^1D)$, $J(NO_2)$, $J(H_2O_2)$, $J(HONO)$, $J(HCHO)$, and $J(NO_3)$) were input into the OBM-MCM model as constraints to realize model simulation localization. Secondly, the model performance of AOI as mentioned in the second question was reasonably acceptable in this study. Hence, the simulated Rchem values could well reflect the actual local atmospheric photochemistry.

The in-situ ozone concentration change is a result of both physical and chemical processes. The $O_3$ concentration change rate (Rmeas) can be determined by the derivative of the observed $O_3$ concentration. The difference between Rmeas and Rchem is caused by physical processes, including horizontal and/or vertical transportation, deposition, and so on, and many studies showed that the impacts of deposition were minor. Anymore, the changes of near-surface winds were corresponding to the variation of the Rtrans in our study. In some relevant studies, their results also suggested that this method can capture the variations in physical processes, thereby, this calculation method could reasonably quantify the contributions of regional transport (Zhang et al., 2021; Xue et al., 2014; Tan et al., 2018; Chen et al., 2020).

About the northerly $O_3$ polluted airflow, we revised this sentence as "the northerly airflow with high $O_3$ or its precursors from an industrial city adjacent to Xiamen of Quanzhou or polluted regions of Yangtze River Delta". Figure R1 shows the 72 h back trajectories in spring and autumn, when the northerly air masses appear frequently in our observation site. In the four pictures of Fig. R1, we could find that the air masses coming from the north carried higher $O_3$ concentration than air masses coming from other directions, attributing to economic and industrially developed areas in the north direction of Xiamen.

About the negative Rtran observed around noontime:
The maximum daily value of $O_3$ at this observation site generally appeared at around 15:00 LT without regional transport, and the values appeared at around 17:00 LT when there was significant regional transport. In Figure 11, we found that the $O_3$ concentrations still showed two peaks at around 15:00 and 17:00 LT, and $O_3$ concentrations rose slowly, or even decreased firstly and then increased between the two peaks. When the $O_3$ concentrations rose slowly or decreased, the Rmeas values would be close to 0 or less than 0, which were less than the Rchem values (Rchem

values were positive until sunset). Under these circumstances, the local photochemical production kept producing $O_3$, while $O_3$ concentrations remained the same or even decreased, which could be attributed to the favorable atmospheric conditions in diluting pollutants ($O_3$ export). In conclusion, the negative Rtran observed around noontime is a phenomenon caused by favorable atmospheric diffusion conditions, which also happened in other regions (Beijing, Shanghai, Guangzhou, Lanzhou, Chengdu and the Yellow River Delta region) (Zhang et al., 2021; Xue et al., 2014; Tan et al., 2018; Chen et al., 2020). The second peak of the Rmeas showing the "M" trend during the daytime was mainly caused by regional transport. And the main revisions in the manuscript are as follows.

"The $O_3$ export was remarkable at around 10:00-16:00, indicating the potential impacts on air quality in downwind areas. Generally, the maximum daily value of $O_3$ at this observation site appeared at around 15:00 LT without regional transport (Wu et al., 2019). In Figure 11, we found that the $O_3$ concentrations showed two peaks at around 15:00 and 17:00 LT, and $O_3$ concentrations rose slowly, or even decreased firstly and then increased between the two peaks. Under these circumstances, the local photochemical production kept producing $O_3$, but the decreased $O_3$ concentrations could be attributed to the favorable atmospheric conditions in diluting pollutants ($O_3$ export)."

[Figure]

**Figure R1. Cluster results of air mass trajectories, relative contributions of $O_3$ concentrations of each air mass by month.**

[Figure]

**Figure 11. O₃ accumulation and contributions from local photochemical production and regional transport, and Rchem, Rtran, and Rmeas in figure caption represent local O₃ photochemical production, regional transport and observed O₃ formation rate, respectively.**

The English may need be improved, e.g.

Line 50, "control factors" to "controlling factors".

Line 53, "destruction rates" to "loss rates".

Line 57, "oxidative" to "oxidation".

etc.

**Response:** We're sorry for the inappropriate expressions. Thanks for your suggestion, and we have invited native speakers in related fields to polish the manuscript.

---

## Author Comment (AC2)

**Response to Reviewers**
Comment on acp-2021-764

**RC2 Anonymous Referee #1**

AOC is key to photochemical reactions and the formation of secondary components like $O_3$ and secondary organic aerosol. This study uses OBM to understand the AOC in a coastal city in China during a typical photochemical episode. It is well organized and suitable for publication in ACP. I have below comments for the authors.

**Response:** Thanks for your valuable comments and positive feedback. We have corrected this manuscript according to your suggestion.

1. OBM is good for understanding the local photochemical formation of $O_3$, but it is not good to evaluate the transport, while back trajectories cannot quantify the contributions. Thus, it is important to show the method of how the regional transport contribution is determined. In this study, the differences between observed $O_3$ changes and local formation were treated as regional transport, which is very misleading. A better method representation should be physical processes instead of regional transport.

**Response:** Thanks for your suggestion, we strongly agree with your suggestions of Rtrans representing physical processes. The in-situ ozone concentration change is a result of both physical and chemical processes. The $O_3$ concentration change rate (Rmeas) can be determined by the observed $O_3$ concentration. The difference between Rmeas and Rchem is caused by physical processes, including horizontal and/or vertical transportation, dry deposition, dilution mixing, and so on, and many studies showed that the impacts of dry deposition were minor. Hence, the differences between observed $O_3$ changes and local formation were mainly caused by the regional transport (note that the effect of atmospheric mixing was also included in this term). Anymore, the changes of near-surface winds were corresponding to the variation of the Rtrans in our study. In some relevant studies, their results also suggested that this method can capture the variations in physical processes, thereby, this calculation method could reasonably quantify the contributions of regional transport at our observation site (Zhang et al., 2021; Xue et al., 2014; Tan et al., 2018; Chen et al., 2020). To avoid misleading, we have revised the relevant content in the manuscript.

"The physical processes (Rtran) were calculated by the equation of Rtran = Rmeas – Rchem, including horizontal and/or vertical transport, dry deposition dilution mixing, and so on. Many studies showed that the impacts of dry deposition were minor, thus the differences between observed $O_3$ changes and local $O_3$ production were mainly caused by the regional transport (note that the effect of atmospheric mixing was also included in this term), which could be treated as regional transport and could reasonably quantify the contributions of regional transport at our observation site (Zhang et al., 2021; Chen et al., 2020)."

2. CO looks very important in OH reactivity, a quick search showed me that it is quite different from other countries, please add comparison or discussion why it is high in this study. (CalNex-LA, BEACHON-SRM08, DISCOVER-AQ)

**Response:** Thanks for your suggestion. About the relatively high fraction of CO in OH reactivity, which was mainly due to the high CO concentrations during the observation period. CO mainly comes from vehicle exhaust and the combustion of fossil fuels. The observation site is a city with high density vehicles, and the high values of observed CO in the morning and evening rush hour also verified the important effects of vehicle emissions. Meanwhile, this pollution event was under the influence of West Pacific Subtropical High, which carries favorable photochemical reaction conditions (high temperature, low RH, and stagnant weather conditions) and promotes the formation and accumulation of pollutants in the southeast coastal area. The relevant contents were discussed in our manuscript, and the main revisions are as follows.

"The high fraction of OVOCs and $NO_2$ in OH reactivity indicated the high aged degree of air mass and the intensive $NO_x$ emissions during the observation period, respectively (Li et al., 2018). However, the fraction of CO to OH reactivity at our observation site was higher than that at an urban site in Los Angeles (Hansen et al., 2021), a rural site in Hong Kong (Li et al., 2018), and a mountain site in Colorado (Nakashima et al., 2014), comparable to that at the urban site of Shanghai (Zhang et al., 2021a), which could be attributed to the abundant CO in our observation site. CO mainly comes from vehicle exhaust and the combustion of fossil fuels, and the observation site is a city with high density vehicles. Meanwhile, this pollution event was under the influence of the WPSH, which promoted the formation and accumulation of pollutants."

3. The episode is just one high $O_3$ event, thus, not necessarily the whole story of $O_3$-NOx-VOCs relationship. It should be cautious when making policy implications.

**Response:** Thanks for your suggestion. As the episode is a typical pollution process in the coastal region, the research results might act as reference for the policy makers. It should be known that is necessary to adjust timely the reduction of VOC and NOx policies as the $O_3$ sensitivity changed under the implementation of control measures. Based on your suggestions, we have revised the relevant content of the manuscript.

"As shown in Fig. 10b, achieving the 5% control target were 1) S(VOCs) is reduced by 15%, while S(NOx) remains unchanged; 2) S(VOCs) is reduced larger than 35%; 3) S(NOx) reduction is higher than 60%. The first scenario of just reducing VOCs emission was the most cost-efficient way for short-term or emergency control of $O_3$. However, NOx, as important precursors of $PM_{2.5}$, need to be reduced according to the long-term multi-pollutant control air quality improvement plan in China, thus the second scenario is a more practical and reasonable way to control air pollution."

"As the episode is a typical pollution process in the coastal region, the research results might act as reference for the policy makers."

4. From Figure 11, the Rtran is mostly opposite to the Rchem, which means local formation and so-called regional transport do not work together to cause high ozone events. The conclusion that "regional transport aggravated the pollution of ozone" is not accurate.

**Response:** Thanks for your suggestion. The regional transport of Rtran was divided into $O_3$ import and $O_3$ export, and the $O_3$ import bringing relatively high $O_3$ concentration caused high ozone events, so we mainly focus on the relationship between positive Rtran and Rchem values. In Figure 11, we found that the $O_3$ concentrations showed two peaks at around 15:00 and 17:00 LT, and $O_3$ concentrations rose slowly, or even decreased firstly and then increased between the two peaks. Under these circumstances, the $O_3$ change rates of Rmeas showed the "M" trend during the daytime. The first transient intense $O_3$ import happened in the early morning (at around 6:00-9:00), leading to a more rapid increase at around 6:00-9:00 LT in $O_3$ concentration than that at 9:00-15:00 LT, when the increase in $O_3$ concentration was mainly due to photochemical reactions. The second $O_3$ import happened at around 15:00-17:00, leading to the second peak of $O_3$ concentration. In conclusion, the first $O_3$ import of regional transport increased the $O_3$ production rate and ozone concentration, and the second $O_3$ import based on the intense photochemical conditions made $O_3$ concentration reaching the maximum peak to exceed the National Ambient Air Quality Standard. Hence, the combined effect of regional transport and local $O_3$ formation led to the pollution event, and the regional transport made the $O_3$ concentration exceeding the standard. We have revised the expression to make the conclusion accurate, and the revisions in the manuscript are as follows.

"The first transient intense $O_3$ import happened in the early morning (at around 6:00-9:00), leading to a rapid increase in $O_3$ concentration, which was mainly attributed to the residual ozone from the day before. The $O_3$ export was remarkable at around 10:00-16:00, indicating the potential impacts on air quality in downwind areas. Generally, the maximum daily value of $O_3$ at this observation site appeared at around 15:00 LT without regional transport (Wu et al., 2019). In Figure 11, we found that the $O_3$ concentrations showed two peaks at around 15:00 and 17:00 LT, and $O_3$ concentrations rose slowly, or even decreased firstly and then increased between the two peaks. Under these circumstances, the local photochemical production kept producing $O_3$, but the decreased $O_3$ concentrations could be attributed to the favorable atmospheric conditions in diluting pollutants ($O_3$ export). When the near-surface wind direction changed from northeast to southeast, the second $O_3$ import phenomenon occurred in the afternoon (16:00-19:00 LT) in four days (20, 25 27 and 29 Sep.). Due to the persistence of Rtran in the afternoon, the daily maximum $O_3$ values appeared at around 17:00 LT."

"Overall, the results indicate that the three conditions of local photochemical production, synoptic situations, and regional transport played very important roles in the pollution event."

5. Some expresses are not in scientific mode, for example, 1) In Abstract, "were the important primary sources of $RO_X$", $O_3$ and HCHO are not emission sources, so not proper to use primary. 2) how the uncertainties are calculated? OH contributed to 91±23%, at what cases, you have a larger than 100% contribution?

**Response:** Thanks for your suggestion. The answers to the questions were shown below.

1) The expression of the primary source of $RO_X$ in our manuscript means chain initial reaction, a reaction that molecules rely on heat and light decomposing into free radicals, and also means the major source. The $RO_X$ chain initial reactions are uniformly expressed as primary sources in related researches (Zhang et al., 2021; Xue et al., 2014; Tan et al., 2018; Chen et al., 2020). For better understanding, I changed the expression as "Photolysis of nitrous acid (HONO, 33±14%), $O_3$ (25±13%), formaldehyde (HCHO, 20±5%), and other OVOCs (17±2%) were major ROx sources, which played initiation roles in atmospheric oxidation processes".

2) About the uncertainties of the model simulation results, the index of agreement (IOA) can be used to judge the reliability of the model simulation results, and its equation is (Liu et al., 2019):

$$IOA = 1 - \frac{\sum_{i=1}^{n}(O_i - S_i)^2}{\sum_{i=1}^{n}(|O_i - \bar{O}| - |S_i - \bar{O}|)^2}$$

where $S_i$ is simulated value, $O_i$ represents observed value, $\bar{O}$ is the average observed values, and n is the sample number. The IOA range is 0-1, and the higher the IOA value is, the better agreement between simulated and observed values is. In many studies, when IOA ranges from 0.68 to 0.89 (Wang et al., 2018a), the simulation results are reasonable, and the IOA in our research is 0.80. Hence, the performance of the OBM-MCM model was reasonably acceptable.

The value of ±23% here represents the standard deviation based on the samples, reflecting the degree of dispersion of the values. During the daytime, the OH contributing to AOC ranged from 13% to 99%, and there was no case where OH contribution was larger than 100%.

**Reference:**

Chen, T., Xue, L., Zheng, P., Zhang, Y., Liu, Y., Sun, J., Han, G., Li, H., Zhang, X., Li,

Y., Li, H., Dong, C., Xu, F., Zhang, Q., and Wang, W.: Volatile organic compounds and ozone air pollution in an oil production region in northern China, Atmospheric Chemistry and Physics, 20, 7069-7086, 10.5194/acp-20-7069-2020, 2020.

Li, Z., Xue, L., Yang, X., Zha, Q., Tham, Y. J., Yan, C., Louie, P. K. K., Luk, C. W. Y., Wang, T., and Wang, W.: Oxidizing capacity of the rural atmosphere in Hong Kong, Southern China, Sci Total Environ, 612, 1114-1122, 10.1016/j.scitotenv.2017.08.310, 2018.

Tan, Z., Lu, K., Jiang, M., Su, R., Dong, H., Zeng, L., Xie, S., Tan, Q., and Zhang, Y.: Exploring ozone pollution in Chengdu, southwestern China: A case study from radical chemistry to $O_3$-VOC-NOx sensitivity, Sci Total Environ, 636, 775-786, 10.1016/j.scitotenv.2018.04.286, 2018.

Xue, L. K., Wang, T., Gao, J., Ding, A. J., Zhou, X. H., Blake, D. R., Wang, X. F., Saunders, S. M., Fan, S. J., Zuo, H. C., Zhang, Q. Z., and Wang, W. X.: Ground-level ozone in four Chinese cities: precursors, regional transport and heterogeneous processes, Atmos. Chem. Phys., 14, 13175-13188, 10.5194/acp-14-13175-2014, 2014.

Wu, X., Xu, L., Hong, Y., Chen, J., Qiu, Y., Hu, B., Hong, Z., Zhang, Y., Liu, T., Chen, Y., Bian, Y., Zhao, G., Chen, J., and Li, M.: The air pollution governed by subtropical high in a coastal city in Southeast China: Formation processes and influencing mechanisms, Sci Total Environ, 692, 1135-1145, 10.1016/j.scitotenv.2019.07.341, 2019.

Liu, X., Lyu, X., Wang, Y., Jiang, F., and Guo, H.: Intercomparison of $O_3$ formation and radical chemistry in the past decade at a suburban site in Hong Kong, Atmos. Chem. Phys., 19, 5127-5145, 10.5194/acp-19-5127-2019, 2019.

Li, Z., Xue, L., Yang, X., Zha, Q., Tham, Y. J., Yan, C., Louie, P. K. K., Luk, C. W. Y., Wang, T., and Wang, W.: Oxidizing capacity of the rural atmosphere in Hong Kong, Southern China, Sci Total Environ, 612, 1114-1122, 10.1016/j.scitotenv.2017.08.310, 2018.

Hansen R F, Griffith S M, Dusanter S, et al.: Measurements of Total OH Reactivity During CalNex-LA. J. Geophys. Res, Atmos, 126(11), e2020JD032988, 2021.

Nakashima Y , Kato S , Greenberg J , et al.: Total OH reactivity measurements in ambient air in a southern Rocky mountain ponderosa pine forest during BEACHON-SRM08 summer campaign. Atmos. Environ., 85(MAR.), 1-8, 2014.

Zhang, K., Duan, Y., Huo, J., Huang, L., Wang, Y., Fu, Q., Wang, Y., and Li, L.: Formation mechanism of HCHO pollution in the suburban Yangtze River Delta region, China: A box model study and policy implementations, Atmos. Environ., 267, 118755, 10.1016/j.atmosenv.2021.118755, 2021a.